# Learning Counterfactually Invariant Predictors

**Francesco Quinzan**[*][†]
*Department of Computer Science, The University of Oxford*

**Cecilia Casolo**[†]
*Techincal University of Munich*
*Helmholtz Munich*
*Munich Center for Machine Learning (MCML)*

**Krikamol Muandet**
*CISPA–Helmholtz Center for Information Security*

**Yucen Luo**
*Citadel Securities*

**Niki Kilbertus**
*Techincal University of Munich*
*Helmholtz Munich*
*Munich Center for Machine Learning (MCML)*

[†] *equal contribution*

**Reviewed on OpenReview:** *https://openreview.net/forum?id=pRt1Vw1DPs*

## Abstract

Notions of counterfactual invariance (CI) have proven essential for predictors that are fair, robust, and generalizable in the real world. We propose graphical criteria that yield a sufficient condition for a predictor to be counterfactually invariant in terms of a conditional independence in the observational distribution. In order to learn such predictors, we propose a model-agnostic framework, called Counterfactually Invariant Prediction (CIP), building on the Hilbert-Schmidt Conditional Independence Criterion (HSCIC), a kernel-based conditional dependence measure. Our experimental results demonstrate the effectiveness of CIP in enforcing counterfactual invariance across various simulated and real-world datasets including scalar and multi-variate settings.

## 1 Introduction

Invariance, or equivariance to certain data transformations, has proven essential in numerous applications of machine learning (ML), since it can lead to better generalization capabilities (Arjovsky et al., 2019; Chen et al., 2020; Bloem-Reddy & Teh, 2020). For instance, in image recognition, predictions ought to remain unchanged under scaling, translation, or rotation of the input image. Data augmentation, an early heuristic to promote such invariances, has become indispensable for successfully training deep neural networks (DNNs) (Shorten & Khoshgoftaar, 2019; Xie et al., 2020). Well-known examples of "invariance by design" include convolutional neural networks (CNNs) for translation invariance (Krizhevsky et al., 2012), group equivariant NNs for general group transformations (Cohen & Welling, 2016), recurrent neural networks (RNNs) and transformers for sequential data (Vaswani et al., 2017), DeepSet (Zaheer et al., 2017) for sets, and graph neural networks (GNNs) for different types of geometric structures (Battaglia et al., 2018).

---

[*]Part of this work was done while Francesco Quinzan visited the Max Planck Institute for Intelligent Systems, Tübingen, Germany.

Many applications in modern ML, however, call for arguably stronger notions of invariance based on causality. This case has been made for image classification, algorithmic fairness (Hardt et al., 2016; Mitchell et al., 2021), robustness (Bühlmann, 2020), and out-of-distribution generalization (Lu et al., 2021). The goal is invariance with respect to hypothetical manipulations of the data generating process (DGP). Various works develop methods that assume observational distributions (across environments or between training and test) to be governed by shared causal mechanisms, but differ due to various types of distribution shifts encoded by the causal model (Peters et al., 2016; Heinze-Deml et al., 2018; Rojas-Carulla et al., 2018; Arjovsky et al., 2019; Bühlmann, 2020; Subbaswamy et al., 2022; Yi et al., 2022; Makar et al., 2022). Typical goals include to train predictors invariant to such shifts, to learn about causal mechanisms and to improve robustness against spurious correlations or out of distribution generalization. The term "counterfactual invariance" has also been used in other out of distribution learning contexts unrelated to our task, e.g., to denote invariance to certain symmetry transformations (Mouli & Ribeiro, 2022).

While we share the broader motivation, these works are orthogonal to ours, because even though counterfactual distributions are also generated from the same causal model, they are fundamentally different from such shifts (Peters et al., 2017). Intuitively, counterfactuals are about events that did not, but could have happened had circumstances been different in a controlled way. A formal discussion of what we mean by counterfactuals is required to properly position our work in the existing literature and describe our contributions. The main contribution of this work is combining a solid theoretical foundation provided in Theorem 3.2 and the flexible, easy-to-use, practical learning framework. Specifically, our work provides a graphical criterion for counterfactual causal inference under specific injectivity conditions, alongside a new model-agnostic learning framework named Counterfactually Invariant Prediction (CIP), which employs a kernel-based conditional dependence measure effective across mixed variable types. We empirically assess CIP in (semi-)synthetic settings and show that it effectively enforces counterfactual invariance.

## 2 Problem setting and related work

### 2.1 Preliminaries and terminology

**Definition 2.1** (Structural causal model (SCM)). *A structural causal model is a tuple $\mathcal{S} = (\mathbf{U}, \mathbf{V}, F, \mathbb{P}_{\mathbf{U}})$ such that $\mathbf{U}$ is a set of background variables that are exogenous to the model; $\mathbf{V}$ is a set of observable (endogenous) variables; $F = \{f_V\}_{V \in \mathbf{V}}$ is a set of functions from (the domains of) $\mathsf{pa}(V) \cup U_V$ to (the domain of) $V$, where $U_V \subset \mathbf{U}$ and $\mathsf{pa}(V) \subseteq \mathbf{V} \setminus \{V\}$ such that $V = f_V(\mathsf{pa}(V), U_V)$; $\mathbb{P}_{\mathbf{U}}$ is a probability distribution over the domain of $\mathbf{U}$. Further, the subsets $\mathsf{pa}(V) \subseteq \mathbf{V} \setminus \{V\}$ are chosen such that the graph $\mathcal{G}$ over $\mathbf{V}$ where the edge $V' \to V$ is in $\mathcal{G}$ if and only if $V' \in \mathsf{pa}(V)$ is a directed acyclic graph (DAG).*

**Observational distribution.** An SCM implies a unique observational distribution over $\mathbf{V}$, which can be thought of as being generated by transforming the distribution over $\mathbb{P}_{\mathbf{U}}$ via the deterministic functions in $F$ iteratively to obtain a distribution over $\mathbf{V}$.[1]

**Interventions.** Given a variable $A \in \mathbf{V}$, an intervention $A \leftarrow a$ amounts to replacing $f_A$ in $F$ with the constant function setting $A$ to $a$. This yields a new SCM, which induces the *interventional distribution* under intervention $A \leftarrow a$.[2] Similarly, we can intervene on multiple variables $\mathbf{V} \supseteq \mathbf{A} \leftarrow \mathbf{a}$. For an outcome (or prediction target) variable $\mathbf{Y} \subset \mathbf{V}$, we then write $\mathbf{Y_a}$ for the outcome in the intervened SCM, also called *potential outcome*. Note that the interventional distribution $\mathbb{P}_{\mathbf{Y_a}}(\mathbf{y})$ differs in general from the conditional distribution $\mathbb{P}_{\mathbf{Y}|\mathbf{A}}(\mathbf{y} \mid \mathbf{a})$.[3] This is typically the case when $Y$ and $A$ have a shared ancestor, i.e., they are confounded. In interventional distributions, potential outcomes are random variables via the exogenous variables $\mathbf{u}$, i.e., $Y_{\mathbf{a}}(u)$ where $\mathbf{u} \sim \mathbb{P}_{\mathbf{U}}$. Hence, interventions capture "population level" properties, i.e., the action is performed for all units $\mathbf{u}$.

**Counterfactuals.** Counterfactuals capture what happens under interventions for a "subset" of possible units $\mathbf{u}$ that are compatible with observations $\mathbf{W} = \mathbf{w}$ for a subset of observed variables $\mathbf{W} \subseteq \mathbf{V}$. This can be described in a three step procedure. (i) *Abduction:* We restrict our attention to units compatible with the

---

[1]Note that all randomness stems from $\mathbb{P}_{\mathbf{U}}$. The observational distribution is well-defined and unique, essentially because every DAG allows for a topological order.

[2]The observational distribution in an intervened SCM is called interventional distribution of the base SCM.

[3]We use $\mathbb{P}$ for distributions (common in the kernel literature) and $\mathbf{Y_a}$ instead of the do notation.

observations, i.e., consider the new SCM $\mathcal{S}^{\mathbf{w}} = (\mathbf{U}, \mathbf{V}, F, \mathbb{P}_{\mathbf{U}|\mathbf{W}=\mathbf{w}})$. (ii) *Intervention:* Within $\mathcal{S}^{\mathbf{w}}$, perform an intervention $\mathbf{A} \leftarrow \mathbf{a}$ on some variables $\mathbf{A}$ (which need not be disjoint from $\mathbf{W}$). (iii) *Prediction:* Finally, we are typically interested in the outcome $\mathbf{Y}$ in an interventional distribution of $\mathcal{S}^{\mathbf{w}}$, which we denote by $\mathbb{P}_{\mathbf{Y}_{\mathbf{a}}^*|\mathbf{W}=\mathbf{w}}(\mathbf{y})$ and call a *counterfactual distribution*: "Given that we have observed $\mathbf{W} = \mathbf{w}$, what would $\mathbf{Y}$ have been had we set $\mathbf{A} \leftarrow \mathbf{a}$, instead of the value $\mathbf{A}$ has actually taken?". In this notation the conditioning set $\mathbf{W}$ is conditioned on in the pre-interventional world. Counterfactuals capture properties of a "subpopulation" $\mathbf{u} \sim \mathbb{P}_{\mathbf{U}|\mathbf{W}=\mathbf{w}}$ compatible with the observations.[4] Even for fine-grained $\mathbf{W}$, there may be multiple units $\mathbf{u}$ in the support of this distribution. In contrast, "unit level counterfactuals" often considered in philosophy contrast $\mathbf{Y}_{\mathbf{a}}^*(\mathbf{u})$ with $\mathbf{Y}_{\mathbf{a}'}^*(\mathbf{u})$ for a single unit $\mathbf{u}$. Such unit level counterfactuals are too fine-grained in our setting. Hence, our used definition of counterfactual invariance is:

**Definition 2.2** (Counterfactual invariance). *Let $\mathbf{A}$, $\mathbf{W}$ be (not necessarily disjoint) sets of nodes in a given SCM. Then, $\mathbf{Y}$ is* counterfactually invariant in $\mathbf{A}$ w.r.t. $\mathbf{W}$ *if* $\mathbb{P}_{\mathbf{Y}_{\mathbf{a}}^*|\mathbf{W}=\mathbf{w}}(\mathbf{y}) = \mathbb{P}_{\mathbf{Y}_{\mathbf{a}'}^*|\mathbf{W}=\mathbf{w}}(\mathbf{y})$ *almost surely, for all $\mathbf{a}, \mathbf{a}'$ in the domain of $\mathbf{A}$ and all $\mathbf{w}$ in the domain of $\mathbf{W}$.*[5]

**Predictors in SCMs.** Ultimately, we aim at learning a predictor $\hat{\mathbf{Y}}$ for the outcome $\mathbf{Y}$. Originally, the predictor $\hat{\mathbf{Y}}$ is not part of the DGP, because we get to learn $f_{\hat{\mathbf{Y}}}$ from data. Using supervised learning, the predictor $f_{\hat{\mathbf{Y}}}$ depends both on the chosen inputs $\mathbf{X} \subset \mathbf{V}$ as well as the target $\mathbf{Y}$. However, once $f_{\hat{\mathbf{Y}}}$ is fixed, it is a deterministic function with arguments $\mathbf{X} \subset \mathbf{V}$, so $(\mathbf{U}, \mathbf{V} \cup \{\hat{\mathbf{Y}}\}, F \cup \{f_{\hat{\mathbf{Y}}}\}, \mathbb{P}_{\mathbf{U}})$ is a valid SCM and we can consider $\hat{\mathbf{Y}}$ an observed variable with incoming arrows from only $\mathbf{X}$. Hence, the definition of counterfactual invariance can be applied to the predictor $\hat{\mathbf{Y}}$.

**Kernel mean embeddings (KME).** Our method relies on kernel mean embeddings (KMEs). We describe the main concepts pertaining KMEs and refer the reader to Smola et al. (2007); Schölkopf et al. (2002); Berlinet & Thomas-Agnan (2011); Muandet et al. (2017) for details. Fix a measurable space $\mathscr{Y}$ with respect to a $\sigma$-algebra $\mathcal{F}_{\mathscr{Y}}$, and consider a probability measure $\mathbb{P}$ on the space $(\mathscr{Y}, \mathcal{F}_{\mathscr{Y}})$. Let $\mathcal{H}$ be a reproducing kernel Hilbert space (RKHS) with a bounded kernel $k_{\mathbf{Y}} \colon \mathscr{Y} \times \mathscr{Y} \to \mathbb{R}$, i.e., $k_{\mathbf{Y}}$ is such that $\sup_{\mathbf{y} \in \mathscr{Y}} k(\mathbf{y}, \mathbf{y}) < \infty$. The kernel mean embedding $\mu_{\mathbb{P}}$ of $\mathbb{P}$ is defined as the expected value of the function $k(\,\cdot\,, \mathbf{y})$ with respect to $\mathbf{y}$, i.e., $\mu_{\mathbb{P}} \coloneqq \mathbb{E}[k(\,\cdot\,, \mathbf{y})]$. The definition of KMEs can be extended to conditional distributions (Fukumizu et al., 2013; Grünewälder et al., 2012; Song et al., 2009; 2013). Consider two random variables $\mathbf{Y}$, $\mathbf{S}$, and denote with $(\Omega_{\mathbf{Y}}, \mathcal{F}_{\mathbf{Y}})$ and $(\Omega_{\mathbf{S}}, \mathcal{F}_{\mathbf{S}})$ the respective measurable spaces. These random variables induce a probability measure $\mathbb{P}_{\mathbf{Y},\mathbf{S}}$ in the product space $\Omega_{\mathbf{Y}} \times \Omega_{\mathbf{S}}$. Let $\mathcal{H}_{\mathbf{Y}}$ be a RKHS with a bounded kernel $k_{\mathbf{Y}}(\cdot, \cdot)$ on $\Omega_{\mathbf{Y}}$. We define the KME of a conditional distribution $\mathbb{P}_{\mathbf{Y}|\mathbf{S}}(\,\cdot\mid \mathbf{s})$ via $\mu_{\mathbf{Y}|\mathbf{S}=\mathbf{s}} \coloneqq \mathbb{E}[k_{\mathbf{Y}}(\,\cdot\,, \mathbf{y}) \mid \mathbf{S} = \mathbf{s}]$. Here, the expected value is taken over $\mathbf{y}$. KMEs of conditional measures can be estimated from samples (Grünewälder et al., 2012). Pogodin et al. (2022) recently proposed an efficient kernel-based regularizer for learning features of input data that allow for estimating a target while being conditionally independent of a distractor given the target. Since CIP ultimately enforces conditional independence (see Theorem 3.2), we believe it could further benefit from leveraging the efficiency and convergence properties of their technique, which we leave for future work.

## 2.2 Related work and contributions

While we focus on counterfactuals in the SCM framework (Pearl, 2000; Peters et al., 2016), there are different incompatible frameworks to describe counterfactuals (von Kügelgen et al., 2022; Dorr, 2016; Woodward, 2021), which may give rise to orthogonal notions of counterfactual invariance.

Research on algorithmic fairness has explored a plethora of causal "invariance" notions with the goal of achieving fair predictors (Loftus et al., 2018; Carey & Wu, 2022; Plecko & Bareinboim, 2022). Kilbertus et al. (2017) conceptually introduce a notion based on group-level interventions, which has been refined to take into account more fine-grained context by Salimi et al. (2019); Galhotra et al. (2022), who then obtain fair predictors by viewing it as a database repair problem or a causal feature selection problem, respectively. A counterfactual-level definition was proposed by Kusner et al. (2017) and followed up by path-specific

---

[4] Note that conditioning in an interventional distribution is different from a counterfactual and our notation is quite subtle here $\mathbb{P}_{\mathbf{Y}_{\mathbf{a}}^*}(\mathbf{y} \mid \mathbf{W} = \mathbf{w}) \neq \mathbb{P}_{\mathbf{Y}_{\mathbf{a}}^*|\mathbf{W}=\mathbf{w}}(\mathbf{y})$.

[5] With an abuse of notation, if $\mathbf{W} = \emptyset$ then the requirement of conditional counterfactual invariance becomes $\mathbb{P}_{\mathbf{Y}_{\mathbf{a}}}(\mathbf{y}) = \mathbb{P}_{\mathbf{Y}_{\mathbf{a}'}}(\mathbf{y})$ almost surely, for all $\mathbf{a}, \mathbf{a}'$ in the domain of $\mathbf{A}$. The "almost surely" part in our definition merely refers to the type of equality of distributions and is not related to the "almost sure" in a.s.-CI defined by Fawkes & Evans (2023).

counterfactual notions (Nabi & Shpitser, 2018; Chiappa, 2019), where the protected attribute may take different values along different paths to the outcome. Recently, Dutta et al. (2021) developed an information theoretic framework to decompose the overall causal influence allowing for exempted variables and properly dealing with synergies across different paths.

Our focus is on counterfactuals because they are fundamentally more expressive than mere interventions (Pearl, 2000; Bareinboim et al., 2022), but do not require a fine-grained path- or variable-level judgment of "allowed" and "disallowed" paths or variables, which may be challenging to devise in practice. Since CI already requires strong assumptions, we leave path-specific counterfactuals—even more challenging in terms of identifiability (Avin et al., 2005)—for future work. While our Definition 2.2 requires equality in distribution, Veitch et al. (2021) suggest a definition of a counterfactually invariant predictor $f_{\hat{\mathbf{Y}}}$ which requires almost sure equality of $\hat{\mathbf{Y}}_{\mathbf{a}}^*$ and $\hat{\mathbf{Y}}_{\mathbf{a}'}^*$, where we view $\hat{\mathbf{Y}}$ as an observed variable in the SCM as described above. Fawkes & Evans (2023) recently carefully formulated various precise technical definitions of what counterfactual invariance may mean in different contexts, such as "almost sure CI", "distributional CI", and "CI of predictors". They provide various connections between them such as the fact that $f_{\hat{\mathbf{Y}}}$ being $\mathcal{F}$-CI is equivalent to $\hat{\mathbf{Y}}$ being $\mathcal{D}$-CI conditioned on $\mathbf{X}$, yielding an equivalence to the definition of counterfactual fairness (Kusner et al., 2017). The notion of counterfactual invariance in Definition 2.2 is most closely related to $\mathcal{D}$-CI from Fawkes & Evans (2023), but we do not enforce conditioning on the intervening variable.

Inspired by problems in natural language processing (NLP), Veitch et al. (2021) aim at "stress-testing" models for spurious correlations. It differs from our work in that they (i) focus only on two specific graphs, and (ii) provide a *necessary* but not sufficient criterion for CI in terms of a conditional independence. Their method enforces the conditional independence via maximum mean discrepancy (MMD) (in *discrete settings only*). However, enforcing a consequence of CI, does not necessarily improve CI. Indeed, Fawkes & Evans (2023, Prop. 4.4) show that while a.s.-CI implies certain conditional independencies, no set of conditional independencies implies any bounds on the difference in counterfactuals. On the contrary, distributional notions of CI such as $\mathcal{D}$-CI or our Definition 2.2 are weaker than a.s.-CI (see (Fawkes & Evans, 2023, Lem. 2.4). Therefore, these weaker notions can indeed be written equivalently as conditional independencies in the observational distribution under additional assumptions about the data generating mechanism (Fawkes & Evans, 2023, Lem. A.3). For example, Fawkes & Evans (2023, Lem. A.3) shows that $\mathcal{D}$-CI can be implied by conditional independence in special settings where the counterfactual distribution is identified from the observational one. Similarly, our reduction of CI (as in Definition 2.2 ) to conditional independence in Theorem 3.2 requires a strong injectivity assumption that essentially amounts to being able to remove exogenous uncertainty.

**Contributions.** We provide such a sufficient graphical criterion for $\mathcal{D}$-CI under an injectivity condition of a structural equation. Depending on the assumed causal graph, this can also come at the cost of requiring certain variables to be observed. As our main contribution, we propose a model-agnostic learning framework, called Counterfactually Invariant Prediction (CIP), using a kernel-based conditional dependence measure that also works for mixed categorical and continuous, multivariate variables. We evaluate CIP extensively in (semi-)synthetic settings and demonstrate its efficacy in enforcing counterfactual invariance even when the strict assumptions may be violated.

## 3 Counterfactually invariant prediction (CIP)

### 3.1 Sufficient criterion for counterfactual invariance

We will now establish a sufficient graphical criterion to express CI as conditional independence in the observational distribution, rendering it estimable from data. First, we need some terminology.

**Graph terminology.** Consider a path $\pi$ (a sequence of distinct adjacent nodes) in a DAG $\mathcal{G}$. A set of nodes $\mathbf{S}$ is said to *block* $\pi$, if $\pi$ contains a triple of consecutive nodes $A, B, C$ such that one of the following hold: (i) $A \to B \to C$ or $A \leftarrow B \leftarrow C$ or $A \leftarrow B \to C$ and $B \in \mathbf{S}$; (ii) $A \to B \leftarrow C$ and neither $B$ nor any descendent of $B$ is in $\mathbf{S}$. Further, we call $\pi$ a *causal path* between sets of nodes $\mathbf{A}, \mathbf{B}$, when it is a directed path from a node in $\mathbf{A}$ to a node in $\mathbf{B}$. A causal path $\pi$ is a *proper causal path* if it only intersects $\mathbf{A}$ at the first node in $\pi$. Finally, we denote with $\mathcal{G}_{\mathbf{A}}$ the graph obtained by removing from $\mathcal{G}$ all incoming arrows

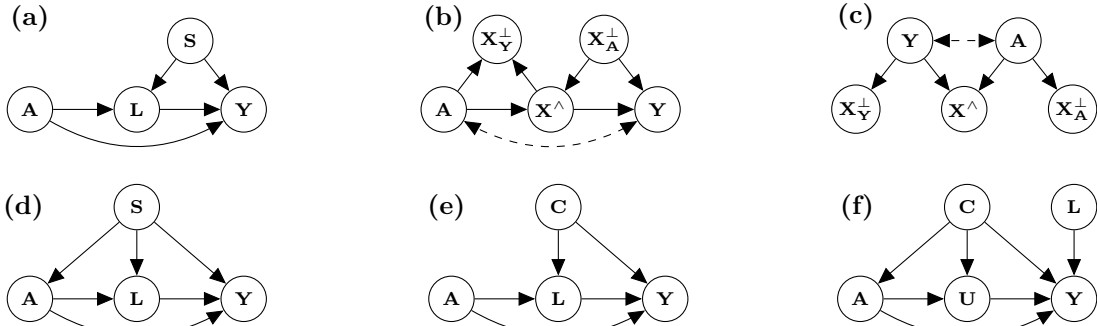

Figure 1: **(a)** Exemplary graph in which a predictor $\hat{\mathbf{Y}}$ with $\hat{\mathbf{Y}} \perp\!\!\!\perp \mathbf{A} \cup \mathbf{L} \mid \mathbf{S}$ is CI in $\mathbf{A}$ w.r.t. $\{\mathbf{L}, \mathbf{S}\}$. **(b)-(c)** Causal and anti-causal structure from Veitch et al. (2021) where $\mathbf{X}_{\mathbf{A}}^{\perp}$ is not causally influenced by $\mathbf{A}$, $\mathbf{X}_{\mathbf{Y}}^{\perp}$ does not causally influence $\mathbf{Y}$, and $\mathbf{X}^{\wedge}$ is both influenced by $\mathbf{A}$ and influences $\mathbf{Y}$. **(d)** Assumed causal structure for the synthetic experiments, see Section 4.1 for details. **(e)** Assumed causal graph for the UCI Adult dataset (Section 4.3), where $\mathbf{A} = \{\text{Gender}, \text{Age}\}$. **(f)** Causal structure for our semi-synthetic image experiments (Section 4.2), where $\mathbf{A} = \{\text{Pos.X}\}$, $\mathbf{U} = \{\text{Scale}\}$, $\mathbf{C} = \{\text{Shape}, \text{Pos.Y}\}$, $\mathbf{L} = \{\text{Color}, \text{Orientation}\}$, and $\mathbf{Y} = \{\text{Outcome}\}$.

into nodes in $\mathbf{A}$. We now define the notion of valid adjustment sets (Shpitser et al., 2010, Def. 5), which our graphical criterion for CI relies on.

**Definition 3.1** (valid adjustment set). *Let $\mathcal{G}$ be a causal graph and let $\mathbf{X}$, $\mathbf{Y}$ be disjoint (sets of) nodes in $\mathcal{G}$. A set of nodes $\mathbf{S}$ is a valid adjustment set for $(\mathbf{X}, \mathbf{Y})$, if (i) no element in $\mathbf{S}$ is a descendant in $\mathcal{G}_{\mathbf{X}}$ of any node $W \notin \mathbf{X}$ which lies on a proper causal path from $\mathbf{X}$ to $\mathbf{Y}$, and (ii) $\mathbf{S}$ blocks all non-causal paths from $\mathbf{X}$ to $\mathbf{Y}$ in $\mathcal{G}$.*

We now state the sufficient graphical criterion that renders CI equivalent to a conditional independence. The proof builds on known results from the literature such as the full characterization of valid adjustment sets in fully observed structural causal models and the backdoor criteria, but the final statement requires a substantial original theoretical contribution.

**Theorem 3.2.** *Let $\mathcal{G}$ be a causal graph, $\mathbf{A}$, $\mathbf{W}$ be two (not necessarily disjoint) sets of nodes in $\mathcal{G}$, such that $(\mathbf{A} \cup \mathbf{W}) \cap \mathbf{Y} = \emptyset$, let $\mathbf{S}$ be a valid adjustment set for $(\mathbf{A} \cup \mathbf{W}, \mathbf{Y})$. Further, for any random variable $X \in \mathbf{W} \setminus \mathbf{A}$ denote with $g_X(\mathsf{pa}(X), U_X)$ its structural equation, and suppose that $\mathsf{pa}(X) \subseteq \mathbf{A} \cup \mathbf{W}$. Suppose that $g_X$ is injective in the variable $\mathbf{u}$.[6] Then, in all SCMs compatible with $\mathcal{G}$, if a predictor $\hat{\mathbf{Y}}$ satisfies $\hat{\mathbf{Y}} \perp\!\!\!\perp \mathbf{A} \cup \mathbf{W} \mid \mathbf{S}$, then $\hat{\mathbf{Y}}$ is counterfactually invariant in $\mathbf{A}$ with respect to $\mathbf{W}$.*

**Assumptions.** First, we do *not* assume the SCM (or any structural equations) to be known. We do assume the causal graph to be known. This is a standard assumption widely made in the causality literature, even though it is a strong one (Cartwright, 2007). The additional assumption of Theorem 3.2, namely injectivity of $g_X$ is satisfied—but more general than—a wide variety of commonly considered models in causality such as the widely used Additive Noise Models (Peters et al., 2017). More broadly, Fawkes & Evans (2023, Lem. A.3) show that to achieve CI from the observational distribution and the causal graph, additional assumptions are always required.

## 3.2 Example use-cases of counterfactually invariant prediction

Fig. 1(a) shows an exemplary graph in which the outcome $\mathbf{Y}$ is affected by (disjoint) sets $\mathbf{A}$ (in which we want to be CI), $\mathbf{L}$ and $\mathbf{S}$ (inputs to $f_{\hat{\mathbf{Y}}}$). We consider $\mathbf{W} = \mathbf{L} \cup \mathbf{A} \cup \mathbf{S}$. Here we aim to achieve $\hat{\mathbf{Y}} \perp\!\!\!\perp \mathbf{A} \cup \mathbf{L} \mid \mathbf{S}$ to obtain CI in $\mathbf{A}$ w.r.t. $\mathbf{W}$. In our synthetic experiments, we also allow $\mathbf{S}$ to affect $\mathbf{A}$, see Fig. 1(d). Let us further illustrate concrete potential applications of CI, which we later also study in our experiments.

---

[6]The injectivity of $g_X$ is defined as follows. Consider two pairs $\{\mathbf{p}, u\}$ and $\{\mathbf{p}, u'\}$ with $\mathbf{p}$ in the support of $\mathsf{pa}(X)$ and $u, u'$ in the support of $U_X$. Suppose that $\mathbb{P}_{\mathsf{pa}(X), U_X}(\mathbf{p}, u) \neq 0$ and $\mathbb{P}_{\mathsf{pa}(X), U_X}(\mathbf{p}, u') \neq 0$. Then, it holds $g(\mathbf{p}, u) = g(\mathbf{p}, u')$ if and only if $u = u'$.

**Counterfactual fairness.** Counterfactual fairness (Kusner et al., 2017) informally challenges a consequential decision: "*Would I have gotten the same outcome had my gender, race, or age been different with all else being equal?*". Here $\mathbf{Y} \subset \mathbf{V}$ denotes the outcome and $\mathbf{A} \subset \mathbf{V} \setminus \mathbf{Y}$ the *protected attributes* such as gender, race, or age—protected under anti-discrimination laws (Barocas & Selbst, 2016)—by $\mathbf{A} \subseteq \mathbf{V} \setminus \mathbf{Y}$. Collecting all remaining observed covariates into $\mathbf{W} := \mathbf{V} \setminus \mathbf{Y}$ counterfactual fairness reduces to counterfactual invariance. In experiments, we build a semi-synthetic DGP assuming the graph in Fig. 1(e) for the UCI adult dataset (Kohavi & Becker, 1996).

**Robustness.** CI serves as a strong notion of robustness in settings such as image classification: "*Would the truck have been classified correctly had it been winter in this situation instead of summer?*" For concrete demonstration, we use the dSprites dataset (Matthey et al., 2017) consisting of simple black and white images of different shapes (squares, ellipses, . . . ), sizes, orientations, and locations. We devise a DGP for this dataset with the graph depicted in Fig. 1(f).

**Text classification.** Veitch et al. (2021) motivate the importance of counterfactual invariance in text classification tasks. Specifically, they consider the causal and anti-causal structures depicted in Veitch et al. (2021, Fig. 1), which we replicate in Fig. 1(b,c). Both diagrams consist of protected attributes $\mathbf{A}$, observed covariates $\mathbf{X}$, and outcomes $\mathbf{Y}$. To apply our sufficient criterion to their settings, we must assume that $\mathbf{A}$ and $\mathbf{Y}$ are unconfounded. We show that CIP still performs on par with Veitch et al. (2021) even when this assumption is violated. Theorem 3.2 provides a sufficient condition for CI (Definition 2.2) in terms of the conditional independence $\hat{\mathbf{Y}} \perp\!\!\!\perp \mathbf{A} \cup \mathbf{W} \mid \mathbf{S}$. We next develop an operator $\text{HSCIC}(\hat{\mathbf{Y}}, \mathbf{A} \cup \mathbf{W} \mid \mathbf{S})$ that is (a) efficiently estimable from data, (b) differentiable, (c) a monotonic measure of conditional dependence, and (d) is zero if and only if $\hat{\mathbf{Y}} \perp\!\!\!\perp \mathbf{A} \cup \mathbf{W} \mid \mathbf{S}$. Hence, it is a practical objective to enforce CI.

### 3.3 HSCIC for conditional independence

Consider two sets of random variables $\mathbf{Y}$ and $\mathbf{A} \cup \mathbf{W}$, and denote with $(\Omega_{\mathbf{Y}}, \mathcal{F}_{\mathbf{Y}})$ and $(\Omega_{\mathbf{A} \cup \mathbf{W}}, \mathcal{F}_{\mathbf{A} \cup \mathbf{W}})$ the respective measurable spaces. Suppose that we are given two RKHSs $\mathcal{H}_{\mathbf{Y}}, \mathcal{H}_{\mathbf{A} \cup \mathbf{W}}$ over the support of $\mathbf{Y}$ and $\mathbf{A} \cup \mathbf{W}$ respectively. The tensor product space $\mathcal{H}_{\mathbf{Y}} \otimes \mathcal{H}_{\mathbf{A} \cup \mathbf{W}}$ is defined as the space of functions of the form $(f \otimes g)(\mathbf{y}, [\mathbf{a}, \mathbf{w}]) := f(\mathbf{y})g([\mathbf{a}, \mathbf{w}])$, for all $f \in \mathcal{H}_{\mathbf{Y}}$ and $g \in \mathcal{H}_{\mathbf{A} \cup \mathbf{W}}$. The tensor product space yields a natural RKHS structure, with kernel $k$ defined by $k(\mathbf{y} \otimes [\mathbf{a}, \mathbf{w}], \mathbf{y}' \otimes [\mathbf{a}', \mathbf{w}']) := k_{\mathbf{Y}}(\mathbf{y}, \mathbf{y}')k_{\mathbf{A} \cup \mathbf{W}}([\mathbf{a}, \mathbf{w}], [\mathbf{a}', \mathbf{w}'])$. We refer the reader to Szabó & Sriperumbudur (2017) for more details on tensor product spaces.

**Definition 3.3** (HSCIC)**.** *For (sets of) random variables $\mathbf{Y}$, $\mathbf{A} \cup \mathbf{W}$, $\mathbf{S}$, the HSCIC between $\mathbf{Y}$ and $\mathbf{A} \cup \mathbf{W}$ given $\mathbf{S}$ is defined as the real-valued random variable $HSCIC(\mathbf{Y}, \mathbf{A} \cup \mathbf{W} \mid \mathbf{S}) = H_{\mathbf{Y}, \mathbf{A} \cup \mathbf{W} \mid \mathbf{S}} \circ \mathbf{S}$ where $H_{\mathbf{Y}, \mathbf{A} \cup \mathbf{W} \mid \mathbf{S}}$ is a real-valued deterministic function, defined as $H_{\mathbf{Y}, \mathbf{A} \cup \mathbf{W} \mid \mathbf{S}}(\mathbf{s}) := \|\mu_{\mathbf{Y}, \mathbf{A} \cup \mathbf{W} \mid \mathbf{S}=\mathbf{s}} - \mu_{\mathbf{Y} \mid \mathbf{S}=\mathbf{s}} \otimes \mu_{\mathbf{A} \cup \mathbf{W} \mid \mathbf{S}=\mathbf{s}}\|$ with $\|\cdot\|$ the norm induced by the inner product of the tensor product space $\mathcal{H}_{\mathbf{X}} \otimes \mathcal{H}_{\mathbf{A} \cup \mathbf{W}}$.*

Our Definition 3.3 is motivated by, but differs slightly from Park & Muandet (2020, Def. 5.3), which relies on the Bochner conditional expected value. While it is functionally equivalent (with the same implementation, see Eq. (2)), ours has the benefit of bypassing some technical assumptions required by Park & Muandet (2020). The HSCIC has the following important property.

**Theorem 3.4** (Theorem 5.4 by Park & Muandet (2020))**.** *If the kernel $k$ of $\mathcal{H}_{\mathbf{X}} \otimes \mathcal{H}_{\mathbf{A} \cup \mathbf{W}}$ is characteristic[7], $HSCIC(\mathbf{Y}, \mathbf{A} \cup \mathbf{W} \mid \mathbf{S}) = 0$ almost surely if and only if $\mathbf{Y} \perp\!\!\!\perp \mathbf{A} \cup \mathbf{W} \mid \mathbf{S}$.*

Because "most interesting" kernels such as the Gaussian and Laplacian kernels are characteristic, and the tensor product of translation-invariant characteristic kernels is characteristic again (Szabó & Sriperumbudur, 2017), this natural assumption is non-restrictive in practice. Combining Theorems 3.2 and 3.4, we can now use HSCIC to reliably achieve counterfactual invariance.

**Corollary 3.5.** *Under the assumptions of Theorem 3.2, if $HSCIC(\hat{\mathbf{Y}}, \mathbf{A} \cup \mathbf{W} \mid \mathbf{S}) = 0$ almost surely, then $\hat{\mathbf{Y}}$ is counterfactually invariant in $\mathbf{A}$ with respect to $\mathbf{W}$.*

In addition, since HSCIC is defined in terms of the MMD (Definition 3.3 and Park & Muandet (2020, Def. 5.3)), it inherits the weak convergence property, i.e., if $\text{HSCIC}(\hat{\mathbf{Y}}, \mathbf{A} \cup \mathbf{W} \mid \mathbf{S})$ converges to zero, then the counterfactual distributions (for different intervention values $\mathbf{a}$) weakly converge to the same distribution. We refer to Simon-Gabriel & Schölkopf (2018); Simon-Gabriel et al. (2020) for a precise characterization. Hence,

---

[7]The tensor product kernel $k$ is characteristic if $\mathbb{P}_{\mathbf{Y}, \mathbf{A} \cup \mathbf{W}} \mapsto \mathbb{E}_{\mathbf{y}, [\mathbf{a}, \mathbf{w}]}[k(\,\cdot\,, \mathbf{y} \otimes [\mathbf{a}, \mathbf{w}])]$ is injective.

as HSCIC decreases, the predictor approaches counterfactual invariance and we need not drive HSCIC all the way to zero to obtain meaningful results.

### 3.4 Learning counterfactually invariant predictors (CIP)

Corollary 3.5 justifies our proposed objective, namely to minimize the following loss

$$\mathcal{L}_{\mathrm{CIP}}(\hat{\mathbf{Y}}) = \mathcal{L}(\hat{\mathbf{Y}}) + \gamma \cdot \mathrm{HSCIC}(\hat{\mathbf{Y}}, \mathbf{A} \cup \mathbf{W} \mid \mathbf{S}), \qquad \text{[CIP loss]} \tag{1}$$

where $\mathcal{L}(\hat{\mathbf{Y}})$ is a task-dependent loss function (e.g., cross-entropy for classification, or mean squared error for regression) and $\gamma \geq 0$ is a parameter that regulates the trade-off between predictive performance and counterfactual invariance.

**The meaning of $\gamma$ and how to choose it.** The second term in Eq. (1) amounts to the additional objective of CI, which is typically at odds with predictive performance within the observational distribution $\mathcal{L}$. In practice, driving HSCIC to zero, i.e., viewing our task as a constrained optimization problem, typically deteriorates predictive performance too much to be useful for prediction—especially in small data settings.[8] As the choice of $\gamma$ amounts to choosing an operating point between predictive performance and CI, it cannot be selected in a data-driven fashion. As different settings call for different tradeoffs, we advocate for employing the following procedure: (i) Train an unconstrained predictor for a base predictive performance (e.g., 92% accuracy or 0.21 MSE). (ii) Fix a tolerance level $\alpha$, indicating the maximally tolerable loss in predictive performance (e.g., at most 5% drop in accuracy or at most 10% increase in MSE). (iii) Perform a log-spaced binary search on $\gamma$ (e.g., on $[10^{-4}, 10^4]$) to find the largest $\gamma$ such that the predictive performance of the resulting predictor achieves predictive performance within the tolerance $\alpha$. A similar search for the optimal value of $\gamma$ can be conducted, when there is a fixed requirement for a maximum tolerance of counterfactual invariance as measured by HSCIC.

**Estimating the HSCIC from samples.** The key benefit of HSCIC as a conditional independence measure is that it does not require parametric assumptions on the underlying probability distributions, and it is applicable for any mixed, multi-dimensional data modalities, as long as we can define positive definite kernels on them. Given $n$ samples $\{(\hat{\mathbf{y}}_i, \mathbf{a}_i, \mathbf{w}_i, \mathbf{s}_i)\}_{i=1}^n$, denote with $\hat{K}_{\hat{\mathbf{Y}}}$ the kernel matrix with entries $[\hat{K}_{\hat{\mathbf{Y}}}]_{i,j} := k_{\hat{\mathbf{Y}}}(\hat{\mathbf{y}}_i, \hat{\mathbf{y}}_j)$, and let $\hat{K}_{\mathbf{A} \cup \mathbf{W}}$ be the kernel matrix for $\mathbf{A} \cup \mathbf{W}$. We estimate the $H_{\hat{\mathbf{Y}}, \mathbf{A} \cup \mathbf{W} \mid \mathbf{S}} \equiv H_{\hat{\mathbf{Y}}, \mathbf{A} \cup \mathbf{W} \mid \mathbf{S}}(\cdot)$ as

$$\begin{aligned}
\hat{H}^2_{\hat{\mathbf{Y}}, \mathbf{A} \cup \mathbf{W} \mid \mathbf{S}} = {} & \hat{w}^T_{\hat{\mathbf{Y}}, \mathbf{A} \cup \mathbf{W} \mid \mathbf{S}} \left( \hat{K}_{\hat{\mathbf{Y}}} \odot \hat{K}_{\mathbf{A} \cup \mathbf{W}} \right) \hat{w}_{\hat{\mathbf{Y}}, \mathbf{A} \cup \mathbf{W} \mid \mathbf{S}} \\
& - 2 \left( \hat{w}^T_{\hat{\mathbf{Y}} \mid \mathbf{S}} \hat{K}_{\mathbf{Y}} \hat{w}_{\hat{\mathbf{Y}}, \mathbf{A} \cup \mathbf{W} \mid \mathbf{S}} \right) \left( \hat{w}^T_{\mathbf{A} \cup \mathbf{W} \mid \mathbf{S}} \hat{K}_{\mathbf{A} \cup \mathbf{W}} \hat{w}_{\hat{\mathbf{Y}}, \mathbf{A} \cup \mathbf{W} \mid \mathbf{S}} \right) \\
& + \left( \hat{w}^T_{\hat{\mathbf{Y}} \mid \mathbf{S}} \hat{K}_{\hat{\mathbf{Y}}} \hat{w}_{\hat{\mathbf{Y}} \mid \mathbf{S}} \right) \left( \hat{w}^T_{\mathbf{A} \cup \mathbf{W} \mid \mathbf{S}} \hat{K}_{\mathbf{A} \cup \mathbf{W}} \hat{w}_{\mathbf{A} \cup \mathbf{W} \mid \mathbf{S}} \right),
\end{aligned} \tag{2}$$

where $\odot$ is element-wise multiplication. The functions $\hat{w}_{\hat{\mathbf{Y}} \mid \mathbf{S}} \equiv \hat{w}_{\hat{\mathbf{Y}} \mid \mathbf{S}}(\cdot)$, $\hat{w}_{\mathbf{A} \cup \mathbf{W} \mid \mathbf{S}} \equiv \hat{w}_{\mathbf{A} \cup \mathbf{W} \mid \mathbf{S}}(\cdot)$, and $\hat{w}_{\hat{\mathbf{Y}}, \mathbf{A} \cup \mathbf{W} \mid \mathbf{S}} \equiv \hat{w}_{\hat{\mathbf{Y}}, \mathbf{A} \cup \mathbf{W} \mid \mathbf{S}}(\cdot)$ are found via kernel ridge regression. Caponnetto & Vito (2007) provide the convergence rates of the estimand $\hat{H}^2_{\hat{\mathbf{Y}}, \mathbf{A} \cup \mathbf{W} \mid \mathbf{S}}$ under mild conditions. In practice, computing the HSCIC approximation by the formula in Eq. (2) can be computationally expensive. To speed it up, we can use random Fourier features to approximate $\hat{K}_{\hat{\mathbf{Y}}}$ and $\hat{K}_{\mathbf{A} \cup \mathbf{W}}$ (Rahimi & Recht, 2007; Avron et al., 2017). We emphasize that Eq. (2) allows us to consistently estimate the HSCIC *from observational i.i.d. samples, without prior knowledge of the counterfactual distributions.*

### 3.5 Measuring counterfactual invariance.

Besides predictive performance, e.g., mean squared error (MSE) for regression or accuracy for classification, our key metric of interest is the level of counterfactual invariance achieved by the predictor $\hat{\mathbf{Y}}$. First, we emphasize again that counterfactual distributions are generally not identified from the observational distribution (i.e., from available data) meaning that *CI is generally untestable in practice* from observational

---

[8]In particular, HSCIC does not regularize an ill-posed problem, i.e., it does not merely break ties between predictors with equal $\mathcal{L}(\hat{\mathbf{Y}})$. Hence it also need not decay to zero as the sample size increases.

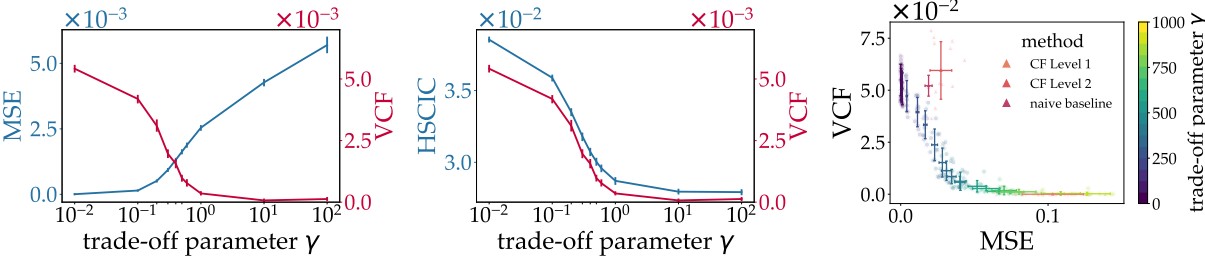

Figure 2: Results on synthetic data. **Left:** trade-off between MSE and counterfactual invariance (VCF). **Middle:** strong correspondence between HSCIC and VCF. **Right:** performance of CIP against baselines CF1 and CF2 and the naive baseline. As $\gamma$ increases, CIP traces out a frontier characterizing the trade-off between MSE and CI. CF2 and the naive baseline are Pareto-dominated by CIP, i.e., we can pick $\gamma$ to outperform CF2 in both MSE and VCF simultaneously. CF1 has zero VCF by design, but worse predictive performance than CIP at near zero VCF. Error bars are standard errors over 10 seeds.

data. We can thus only evaluate CI in (semi-)synthetic settings where we have access to the full SCM and thus all counterfactual distributions.

A measure for CI must capture how the distribution of $\hat{\mathbf{Y}}^*_{\mathbf{a}'}$ changes for different values of $\mathbf{a}'$ across all conditioning values $\mathbf{w}$ (which may include an observed value $\mathbf{A} = \mathbf{a}$). We propose the **V**ariance of **C**ounter**F**actuals (VCF) as a metric of CI

$$\mathrm{VCF}(\hat{\mathbf{Y}}) := \mathbb{E}_{\mathbf{w} \sim \mathbb{P}_{\mathbf{W}}} \left[ \mathrm{var}_{\mathbf{a}' \sim \mathbb{P}_{\mathbf{A}}} \left[ \mathbb{E}_{\hat{\mathbf{Y}}^*_{\mathbf{a}'} | \mathbf{W} = \mathbf{w}}[\hat{\mathbf{y}}] \right] \right]. \tag{3}$$

That is, we quantify how the average outcome varies with the interventional value $\mathbf{a}'$ at conditioning value $\mathbf{w}$ and average this variance over $\mathbf{w}$. For deterministic predictors (point estimators), which we use in all our experiments, the prediction is a fixed value for each input $\mathbb{E}_{\hat{\mathbf{Y}}^*_{\mathbf{a}'} | \mathbf{W} = \mathbf{w}}[\hat{\mathbf{y}}] = \hat{\mathbf{y}})$ and we can drop the inner expectation of Eq. (3). In this case, the variance term in Eq. (3) is zero if and only if $\mathbb{P}_{\hat{\mathbf{Y}}^*_{\mathbf{a}} | \mathbf{W} = \mathbf{w}}(\mathbf{y}) = \mathbb{P}_{\hat{\mathbf{Y}}^*_{\mathbf{a}'} | \mathbf{W} = \mathbf{w}}(\mathbf{y})$ almost surely. Since the variance is non-negative, the outer expectation is zero if and only if the variance term is zero almost surely, yielding the following result.

**Corollary 3.6.** *For point-estimators, $\hat{\mathbf{Y}}$ is counterfactually invariant in $\mathbf{A}$ w.r.t. $\mathbf{W}$ if and only if $VCF(\hat{\mathbf{Y}}) = 0$ almost surely.*

Estimating VCF in practice requires access to the DGP to generate counterfactuals. Given $d$ i.i.d. examples $(\mathbf{w}_i)_{i=1}^d$ from a fixed observational dataset we sample $k$ intervention values from the marginal $\mathbb{P}_{\mathbf{A}}$ and compute corresponding predictions. The inner expectation is simply the deterministic predictor output, and we compute the empirical expectation over the $d$ observed $\mathbf{w}$ values and empirical variances over the $k$ sampled interventional values (for each $\mathbf{w}$). Since the required counterfactuals are by their very nature unavailable in practice, our analysis of VCF is limited to (semi-)synthetic settings. Notably, the proposed procedure for choosing $\gamma$ does not require VCF. Our experiments corroborate the weak convergence property of HSCIC—small HSCIC implies small VCF. Hence, HSCIC may serve as a strong proxy for VCF and thus CI in practice.

In practical terms, HSCIC is considered the relevant metric, which is also estimable from data. Theoretically, however, counterfactual invariance is only implied by an exact HSCIC value of zero (Corollary 3.5). Even if HSCIC exhibits continuity, meaning it approaches zero as distributions converge weakly, VCF may still be preferable for directly assessing counterfactual invariance because it provides a more interpretable metric. Our experimental investigations into VCF aim to establish the practical utility of HSCIC as a measure of counterfactual invariance.

## 4 Experiments

**Baselines.** As many existing methods focus on cruder purely observational or interventional invariances (see Section 2.2), our choice of baselines for true counterfactual invariance is highly limited. First, we compare

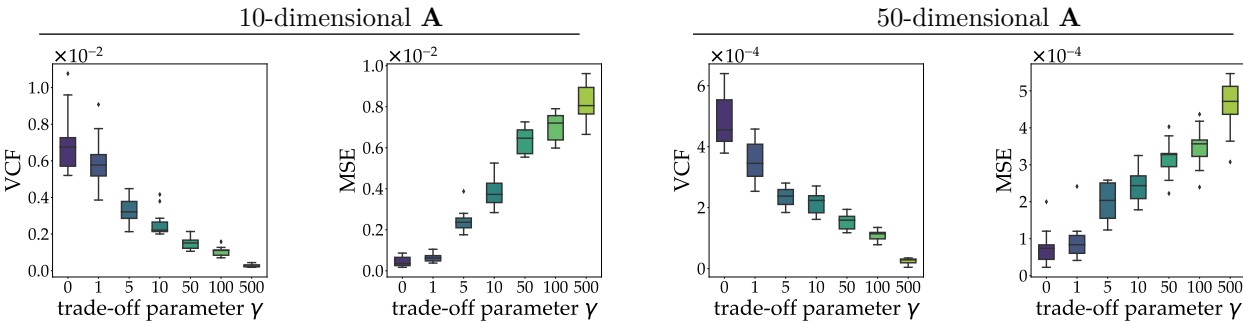

Figure 3: MSE and VCF for synthetic data with 10- and 50-dimensional $\mathbf{A}$ for different $\gamma$ and 15 random seeds per box. CIP reliably achieves CI as $\gamma$ increases.

CIP to Veitch et al. (2021) in their two limited settings, showing that our method performs on par with theirs. Next, we compare to two methods proposed by Kusner et al. (2017) in settings where they apply. CF1 (their 'Level 1') consists of only using non-descendants of $\mathbf{A}$ as inputs to $f_{\hat{\mathbf{Y}}}$. CF2 (their 'Level 2') assumes an additive noise model and uses the residuals of descendants of $\mathbf{A}$ after regression on $\mathbf{A}$ together with non-descendants of $\mathbf{A}$ as inputs to $f_{\hat{\mathbf{Y}}}$. We refer to these two baselines as CF1 and CF2 respectively. We also compare CIP to the 'naive baseline' which consists in training a predictor ignoring $\mathbf{A}$. In settings where $\mathbf{A}$ is binary, we also compare to Chiappa (2019), devised for path-wise counterfactual fairness. Finally, we develop heuristics based on data augmentation as further possible baselines.

### 4.1 Synthetic experiments

First, we generate various synthetic datasets following the causal graph in Fig. 1(d). They contain (i) the prediction targets $\mathbf{Y}$, (ii) variable(s) we want to be CI in $\mathbf{A}$, (iii) covariates $\mathbf{L}$ mediating effects from $\mathbf{A}$ on $\mathbf{Y}$, and (iv) confounding variables $\mathbf{S}$. The goal is to learn a predictor $\hat{\mathbf{Y}}$ that is CI in $\mathbf{A}$ w.r.t. $\mathbf{W} := \mathbf{A} \cup \mathbf{L} \cup \mathbf{S}$. The datasets cover different dimensions for the observed variables and their correlations.

**Model choices and parameters.** For all synthetic experiments, we train fully connected neural networks (MLPs) with MSE loss $\mathcal{L}_{\text{MSE}}(\hat{\mathbf{Y}})$ as the predictive loss $\mathcal{L}$ in Eq. (1) for continuous outcomes $\mathbf{Y}$. We generate 10k samples from the observational distribution in each setting and use an 80 to 20 train-test split. All metrics reported are on the test set. We perform hyper-parameter tuning for MLP hyperparameters based on a random strategy. The HSCIC$(\hat{\mathbf{Y}}, \mathbf{A} \cup \mathbf{W} \mid \mathbf{S})$ term is computed as in Eq. (2) using a Gaussian kernel with amplitude 1.0 and length scale 0.1. The regularization parameter $\lambda$ for the ridge regression coefficients is set to $\lambda = 0.01$. We set $d = 1000$ and $k = 500$ in the estimation of VCF.

**Model performance.** We first study the effect of the HSCIC on accuracy and counterfactual invariance on the simulated dataset. Fig. 2 (left) depicts the expected trade-off between MSE and VCF for varying $\gamma$, whereas Fig. 2 (middle) highlights that HSCIC (estimable from observational data) is a strong proxy of counterfactual invariance measured by VCF (see discussion after Eq. (3)).

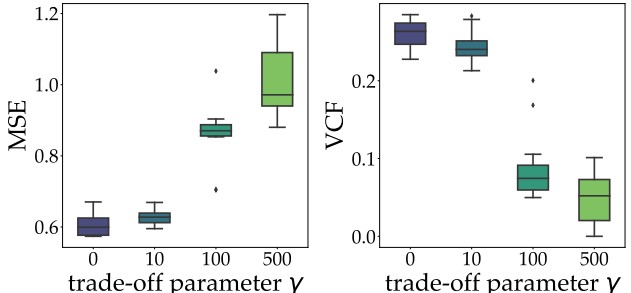

Figure 4: On the dSprites image dataset, CIP trades off MSE for VCF and achieves almost full CI as $\gamma$ increases. Boxes are for 8 random seeds.

Figure 2 (right) compares CIP to baselines for a simulated non-additive noise model. For a suitable choice of $\gamma$, CIP outperforms the baseline CF2 and the naive baseline in both MSE and VCF simultaneously. While CF1 achieves perfect CI by construction (VCF = 0), its MSE is higher than CIP at almost perfect CI (VCF near zero). To conclude, our method flexibly and reliably trades predictive performance for counterfactual invariance via a single parameter $\gamma$ and Pareto-dominates existing methods.

**Effect of dimensionality of A.** A key advantage of CIP is that it can deal with multi-dimensional **A**. We consider simulated datasets, where we gradually increase the dimension of **A**. The results in Fig. 3 for different trade-off parameters $\gamma$ and different dimensions of **A** demonstrate that CIP effectively enforces CI also for multi-dimensional **A**.[9]

## 4.2 Image experiments

We consider an image classification task on the dSprites dataset (Matthey et al., 2017), with a causal model as depicted in Fig. 1(f). This experiment is particularly challenging due to the mixed categorical and continuous variables in **C** (shape, y-pos) and **L** (color, orientation), with continuous **A** (x-pos). We seek a predictor $\hat{\mathbf{Y}}$ that is CI in the x-position w.r.t. all other observed variables. Following Theorem 3.2, we achieve $\hat{\mathbf{Y}} \perp\!\!\!\perp \{\texttt{x-pos}, \texttt{scale}, \texttt{color}, \texttt{orientation}\} \mid \{\texttt{shape}, \texttt{y-pos}\}$ via the HSCIC operator. To accommodate the mixed input types, we first extract features from the images via a CNN and from other inputs via an MLP. We then use an MLP on all concatenated features for $\hat{\mathbf{Y}}$. Fig. 4 shows that CIP gradually enforces CI as $\gamma$ increases and illustrates the inevitable increase of MSE.

## 4.3 Fairness with continuous protected attributes

Finally, we apply CIP to the widely-used UCI Adult dataset (Kohavi & Becker, 1996) and compare it against a 'naive baseline' which simply ignores **A**, CF1, CF2, and path-specific counterfactual fairness (PSCF) (Chiappa, 2019). Like CF1, PSCF always achieves VCF = 0 by design.

We explicitly acknowledge the shortcomings of the UCI Adult dataset to reason about social justice (Ding et al., 2021). Instead, we chose it due to previous investigations into plausible causal structures based on domain knowledge (Zhang et al., 2017). The task is to predict whether an individual's income is above a threshold based on demographic information, including protected attributes. We follow Nabi & Shpitser (2018); Chiappa (2019), where a causal structure is assumed for a subset of the variables as in Fig. 1(e). We choose gender and age as the protected attributes **A**, collect marital status, level of education, occupation, working hours per week, and work class into **L**, and combine the remaining observed attributes in **C**. Our aim is to learn

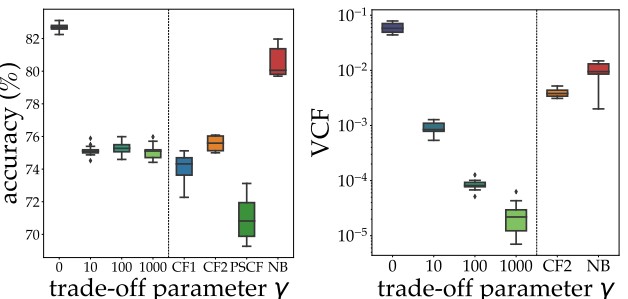

Figure 5: Accuracy and VCF on the Adult dataset. CIP achieves better VCF than CF2 and the naive baseline (NB), improved in accuracy compared to PSCF and is on par with CF1 in accuracy at VCF $\approx 0$.

a predictor $\hat{\mathbf{Y}}$ that is CI in **A** w.r.t. $\mathbf{W} = \mathbf{C} \cup \mathbf{L}$. Achieving (causal) fairness for (mixed categorical and) continuous protected attributes is under active investigation (Mary et al., 2019; Chiappa & Pacchiano, 2021), but directly supported by CIP.

We use an MLP with binary cross-entropy loss for $\hat{\mathbf{Y}}$. Since this experiment is based on real data, the true counterfactual distribution cannot be known. Following Chiappa & Pacchiano (2021) we estimate a possible SCM by inferring the posterior distribution over the unobserved variables using variational autoencoders (Kingma & Welling, 2014). Even though CF1, PSCF always achieves VCF = 0 by design, Figure 5 shows that CIP gradually achieves CI and even manages to keep a constant accuracy after an initial drop. It outperforms PSCF in accuracy and reaches comparable accuracy to CF1 when reaching VCF $\approx 0$. It Pareto-dominates CF2 and can achieve better VCF than the naive baseline, even though NB is more accurate for $\gamma \geq 5$ than CIP.

## 5 Discussion and future work

We developed CIP, a method to learn counterfactually invariant predictors $\hat{\mathbf{Y}}$. First, we presented a sufficient graphical criterion to characterize counterfactual invariance and reduced it to conditional independence in the observational distribution under an injectivity assumption of a causal mechanism. We then built on

---

[9]In all boxplots, boxes represent the interquartile range, the horizontal line is the median, and whiskers show minimum and maximum values, excluding outliers (dots).

kernel mean embeddings and the Hilbert-Schmidt Conditional Independence Criterion to devise an efficiently estimable, differentiable, model-agnostic objective to train CI predictors for mixed continuous/categorical, multi-dimensional variables. We demonstrated the efficacy of CIP in extensive empirical evaluations on various regression and classification tasks.

A limitation of this work is the computational cost of estimating HSCIC, which may limit the scalability of CIP to very high-dimensional settings even when using efficient random Fourier features. While the increased robustness of counterfactually invariant predictors are certainly desirable in many contexts, this presupposes the validity of our assumptions. Thus, an important direction for future work is to assess the sensitivity of CIP to misspecifications of the causal graph or insufficient knowledge of the required adjustment set. Lastly, we envision our graphical criterion and KME-based objective to be useful also for causal representation learning to isolate causally relevant, autonomous factors underlying the data.

Another key limitation of our work, shared by all studies in this domain, is the assumption that the causal graph is known. Assessing the validity of a causal DAG in real-world settings is challenging, since one can often conjecture plausible confounding mechanisms for any missing edge. For this reason, in this work we found it challenging to provide convincing examples that corroborate the "potential validity" of our assumptions. However, our empirical results demonstrate that CIP is effective even when these assumptions are violated. Similarly, our real-world experiments show a comparable invariance and accuracy trade-off trend as the simulated experiments. We believe this to be a better indicator of validity and robustness than isolated toy examples where assumptions may be satisfied. Practitioners using our proposed framework are recommended to take these considerations into account before they apply the method to their own data, and combine causal discovery methods with domain knowledge to assess the validity of the underlying DGP.

## Acknowledgements

Francesco Quinzan acknowledges funding from ELSA: European Lighthouse on Secure and Safe AI project (grant agreement No. 101070617 under UK guarantee). Cecilia Casolo is supported by the DAAD programme Konrad Zuse Schools of Excellence in Artificial Intelligence, sponsored by the Federal Ministry of Education and Research.

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
