## A  Discussion on Related Work

In this section, we give an overview of the definitions used in the related work Fawkes & Evans (2023).

**Definition A.1** (Def. 2.1 by Fawkes et al. (2022)). *A variable $Y$ satisfies **almost sure counterfactual invariance** (a.s.-CI) with respect to A if:*

$$Y_a = Y_{a'} \text{ a.s. for all } a, a'$$

**Definition A.2** (Def. 2.3 by Fawkes & Evans (2023) and Def. 1.1 Veitch et al. (2021)). *A function $f : \mathcal{X} \to \mathcal{Y}$ is **counterfactually invariant** ($\mathcal{F}$-CI) if the variable $\hat{Y} := f(X)$ satisfies almost sure counterfactual invariance. That is:*

$$f(X_a) = f(X_{a'}) \text{ a.s.}$$

**Definition A.3** (Def. 2.2 by Fawkes & Evans (2023)). *A variable $Y$ satisfies **distributional counterfactually invariance** ($\mathcal{D}$-CI) conditional on some set of variables $W$, with respect to A , if:*

$$P(Y_a = y | W = w, A = a) = P(Y_{a'} = y | W = w, A = a)$$

*for all $a$, $a'$ and for almost all $y$, $w$.*

Definition A.3 is similar to the one provided in the main paper Definition 2.2, however enforces conditioning on the intervening variable $A$. As it can be noticed from the definitions, distributional notions of Counterfactual Invariance (CI), such as Definition A.3 and Definition 2.2, are less stringent than Definition A.1, as indicated in the following result in Fawkes & Evans (2023, Lem. 2.4):

**Lemma A.4** (Lem. 2.4 by Fawkes & Evans (2023)). *We have that a.s.-CI implies $\mathcal{D}$-CI conditional on any set of variables, but $\mathcal{D}$-CI implies a.s.-CI only if the conditioning set contains the outcome $Y$.*

## B  Proof of Theorem 3.2

To provide a mostly self-contained proof, we start by stating the necessary definitions and repeat required known results from the literature. Appendices B.1 to B.3 contain these preliminary definitions and results, whereas the final statement and proof provided in Appendix B.4 is our original contribution (making use of the preliminaries stated before).

### B.1  Overview of the proof techniques

We restate the main theorem for completeness.

**Theorem 3.2.** *Let $\mathcal{G}$ be a causal graph, $\mathbf{A}$, $\mathbf{W}$ be two (not necessarily disjoint) sets of nodes in $\mathcal{G}$, such that $(\mathbf{A} \cup \mathbf{W}) \cap \mathbf{Y} = \emptyset$, let $\mathbf{S}$ be a valid adjustment set for $(\mathbf{A} \cup \mathbf{W}, \mathbf{Y})$. Further, for any random variable $X \in \mathbf{W} \setminus \mathbf{A}$ denote with $g_X(\mathsf{pa}(X), U_X)$ its structural equation, and suppose that $\mathsf{pa}(X) \subseteq \mathbf{A} \cup \mathbf{W}$. Suppose that $g_X$ is injective in the variable $\mathbf{u}$.[10] Then, in all SCMs compatible with $\mathcal{G}$, if a predictor $\hat{\mathbf{Y}}$ satisfies $\hat{\mathbf{Y}} \perp\!\!\!\perp \mathbf{A} \cup \mathbf{W} \mid \mathbf{S}$, then $\hat{\mathbf{Y}}$ is counterfactually invariant in $\mathbf{A}$ with respect to $\mathbf{W}$.*

Our proof technique generalizes the work of Shpitser & Pearl (2009). To understand the proof technique, note that conditional counterfactual distributions of the form $\mathbb{P}_{\mathbf{Y}_{\mathbf{a}}^* | \mathbf{W}}(\mathbf{y} \mid \mathbf{w})$ involve quantities *from two different worlds*. The variables $\mathbf{W}$ belong to the pre-interventional world, and the interventional variable $\mathbf{Y_a}$ belongs to the world after performing the intervention $\mathbf{A} \leftarrow \mathbf{a}$. The variable $\mathbf{Y}_{\mathbf{a}|\mathbf{W}}^*$ refers to the counterfactual of $\mathbf{Y_a}$ after conditioning in the pre-interventional world on $\mathbf{W}$.

### B.2  Identifiability of counterfactual distributions

In this section, we discuss the well-known backdoor criterion for the identifiability of conditional distributions, which we will then use to prove Theorem 3.2. To this end, we use the notions of a blocked path and valid adjustment set, which we restate for clarity.

**Definition B.1.** *Consider a path $\pi$ of causal graph $\mathcal{G}$. A set of nodes $\mathbf{Z}$ blocks $\pi$, if $\pi$ contains a triple of consecutive nodes connected in one of the following ways: $N_i \to Z \to N_j$, $N_i \leftarrow Z \to N_j$, with $N_i, N_j \notin \mathbf{Z}$, $Z \in \mathbf{Z}$, or $N_i \to M \leftarrow N_j$ and neither $M$ nor any descendent of $M$ is in $\mathbf{Z}$.*

---

[10]The injectivity of $g_X$ is defined as follows. Consider two pairs $\{\mathbf{p}, u\}$ and $\{\mathbf{p}, u'\}$ with $\mathbf{p}$ in the support of $\mathsf{pa}(X)$ and $u, u'$ in the support of $U_X$. Suppose that $\mathbb{P}_{\mathsf{pa}(X), U_X}(\mathbf{p}, u) \neq 0$ and $\mathbb{P}_{\mathsf{pa}(X), U_X}(\mathbf{p}, u') \neq 0$. Then, it holds $g(\mathbf{p}, u) = g(\mathbf{p}, u')$ if and only if $u = u'$.

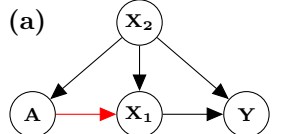 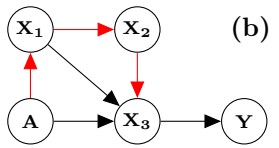 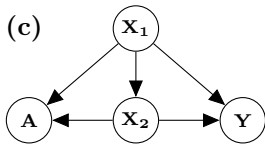

Figure 6: (a) In this causal graph the elements of the set $\mathbf{X} = \{\mathbf{X_1}, \mathbf{X_2}\}$ have depth $\mathsf{depth}(\mathbf{X_1}) = 1$ and $\mathsf{depth}(\mathbf{X_2}) = 0$. (b) In this causal graph the elements of the set $\mathbf{X} = \{\mathbf{X_1}, \mathbf{X_2}, \mathbf{X_3}\}$ have depth $\mathsf{depth}(\mathbf{X_1}) = 1$, $\mathsf{depth}(\mathbf{X_2}) = 2$, $\mathsf{depth}(\mathbf{X_3}) = 3$. (c) In this causal graph the elements of the set $\mathbf{X} = \{\mathbf{X_1}, \mathbf{X_2}\}$ have depth $\mathsf{depth}(\mathbf{X_1}) = \mathsf{depth}(\mathbf{X_2}) = 0$.

Using this definition, we define the concept of a valid adjustment set.

**Definition 3.1** (valid adjustment set). *Let $\mathcal{G}$ be a causal graph and let $\mathbf{X}$, $\mathbf{Y}$ be disjoint (sets of) nodes in $\mathcal{G}$. A set of nodes $\mathbf{S}$ is a valid adjustment set for $(\mathbf{X}, \mathbf{Y})$, if (i) no element in $\mathbf{S}$ is a descendant in $\mathcal{G}_{\mathbf{X}}$ of any node $W \notin \mathbf{X}$ which lies on a proper causal path from $\mathbf{X}$ to $\mathbf{Y}$, and (ii) $\mathbf{S}$ blocks all non-causal paths from $\mathbf{X}$ to $\mathbf{Y}$ in $\mathcal{G}$.*

Definition 3.1 is a useful graphical criterion for the identifiability of counterfactual distributions. In fact, the following theorem holds:

**Theorem B.2** (Theorem 4 by Shpitser et al. (2010)). *Let $\mathcal{G}$ be a causal graph, $\mathbf{A}$, $\mathbf{W}$ be two (not necessarily disjoint) sets of nodes in $\mathcal{G}$, such that $(\mathbf{A} \cup \mathbf{W}) \cap \mathbf{Y} = \emptyset$. Define the set $\mathbf{X} = \mathbf{W} \setminus \mathbf{A}$. Suppose that a set of nodes $\mathbf{S}$ satisfies the adjustment criterion relative to $(\mathbf{A} \cup \mathbf{W}, \mathbf{Y})$ in $\mathcal{G}$. Then, it holds $\mathbf{Y}^*_{\mathbf{a}', \mathbf{x}'} \perp\!\!\!\perp \mathbf{A}, \mathbf{X} \mid \mathbf{S}$ for any intervention $\mathbf{A}, \mathbf{X} \leftarrow \mathbf{a}', \mathbf{x}'$.*

### B.3 $d$-separation and conditional independence

In this section, we discuss a well-known criterion for conditional independence, which we will then use to prove Theorem 3.2. We use the notion of a blocked path, as in Definition B.3 and the concept of $d$-separation as follows.

**Definition B.3** ($d$-Separation). *Consider a causal graph $\mathcal{G}$. Two sets of nodes $\mathbf{X}$ and $\mathbf{Y}$ of $\mathcal{G}$ are said to be $d$-separated by a third set $\mathbf{S}$ if every path from any node of $\mathbf{X}$ to any node of $\mathbf{Y}$ is blocked by $\mathbf{S}$.*

We use the notation $\mathbf{X} \perp\!\!\!\perp_{\mathcal{G}} \mathbf{Y} \mid \mathbf{S}$ to indicate that $\mathbf{X}$ and $\mathbf{Y}$ are $d$-separated by $\mathbf{S}$ in $\mathcal{G}$. We use Definition B.3 as a graphical criterion for conditional independence (Pearl, 2000).

**Lemma B.4** (Markov Property). *Consider a causal graph $\mathcal{G}$, and suppose that two sets of nodes $\mathbf{X}$ and $\mathbf{Y}$ of $\mathcal{G}$ are $d$-separated by $\mathbf{S}$. Then, $\mathbf{X}$ is independent of $\mathbf{Y}$ given $\mathbf{S}$ in any model induced by the graph $\mathcal{G}$.*

The Markov Property is also referred to as $d$-separation property. We use the notation $\mathbf{X} \perp\!\!\!\perp_{\mathcal{G}} \mathbf{Y} \mid \mathbf{S}$ to indicate that $\mathbf{X}$ and $\mathbf{Y}$ are $d$-separated by $\mathbf{S}$ in $\mathcal{G}$.

### B.4 Proof of Theorem 3.2

Before discussing the proof of Theorem 3.2, we prove an additional auxiliary result. All the following theoretical results are novel and based on the previous sections.

**Lemma B.5.** *Let $\mathcal{G}$ be a causal graph, $\mathbf{A}$, $\mathbf{X}$ be two disjoint sets of nodes in $\mathcal{G}$. Suppose that any r.v. $X \subseteq \mathbf{X}$ is defined by a structural equation of the form $X = g(\mathsf{pa}(X), \mathbf{U_X})$, with $\mathsf{pa}(X) \subseteq \mathbf{X} \cup \mathbf{A}$ and $\mathbf{U_X}$ exogenous independent noise. Furthermore, suppose that $g$ is injective in $\mathbf{U_X}$. Then, for any observational values $\mathbf{a}, \mathbf{x}$ with $\mathbb{P}_{\mathbf{X}, \mathbf{A}}(\mathbf{x}, \mathbf{a}) \neq 0$, there exist a value $\mathbf{u}$ in the support of $\mathbf{U_X}$ such that $\mathbb{P}_{\mathbf{U_X} \mid \mathbf{A}=\mathbf{a}, \mathbf{X}=\mathbf{x}}(\mathbf{u}) = \mathbb{1}_{\mathbf{u}}$. Here, $\mathbb{1}_{\mathbf{u}}$ is the Dirac $\delta$-distribution.*

*Proof.* In this proof, we use the following additional notation. Consider a set of nodes $\mathbf{T} \subseteq \mathbf{A} \cup \mathbf{X}$, and let $\mathbf{a}$ be a point in the support of $\mathbf{A}$. We denote with $\mathbf{a_T}$ the restriction of $\mathbf{a}$ to $\mathbf{A} \cap \mathbf{T}$. Similarly, for a point $\mathbf{x}$ in the support of $\mathbf{X}$, we denote with $\mathbf{x_T}$ the restriction of $\mathbf{a}$ to $\mathbf{X} \cap \mathbf{T}$. Furthermore, given two points $\mathbf{a}, \mathbf{x}$ as above, we define $\mathbf{a} \cup \mathbf{x}$ as the only point in the support of $\mathbf{A} \cup \mathbf{X}$ such that $(\mathbf{a} \cup \mathbf{x}) \mid_{\mathbf{A}} = \mathbf{a}$ and $(\mathbf{a} \cup \mathbf{x}) \mid_{\mathbf{X}} = \mathbf{x}$ respectively.

Furthermore, we denote with $\mathbf{X} = g_{\mathbf{X}}(\mathsf{pa}(\mathbf{X}), \mathbf{U_X})$ the structural equation for the joint random variable $\mathbf{X}$. In this equation, $\mathbf{U_X}$ corresponds to the joint exogenous noise. Note that the function $g_X(\mathsf{pa}(X), \mathbf{U_X})$ is

injective in $\mathbf{U_X}$, by the injectivity assumption on the structural equation $g_X(\mathsf{pa}(X), \mathbf{U_X})$ for each $X \subseteq \mathbf{X}$. Define the function $h(\mathbf{p})$ over the support of $\mathsf{pa}(\mathbf{X})$ as

$$h(\mathbf{a}, \mathbf{x}) := \begin{cases} \mathbf{u} \text{ such that } g_\mathbf{X}((\mathbf{a} \cup \mathbf{x})\,|_{\mathsf{pa}(\mathbf{X})}, \mathbf{u}) = \mathbf{x} & \text{if } \mathbb{P}_{\mathsf{pa}(\mathbf{X}),\mathbf{U_X}}((\mathbf{a} \cup \mathbf{x})\,|_{\mathsf{pa}(\mathbf{X})}, \mathbf{u}) \neq 0 \\ 0 & \text{otherwise} \end{cases}$$

Note that by the injectivity of $g_\mathbf{X}$, the function $h$ as above is well-defined. Since $\mathsf{pa}(\mathbf{X}) \subseteq \mathbf{X} \cup \mathbf{A}$, then it holds $\mathbb{P}_{\mathbf{U_X}|\mathbf{A}=\mathbf{a},\mathbf{X}=\mathbf{x}}(\mathbf{u}) = \mathbb{1}_{h(\mathbf{a},\mathbf{x})}$. The claim follows, by defining $\mathbf{u} := h(\mathbf{a}, \mathbf{x})$. □

We use lemma B.5 to prove the following result.
**Lemma B.6.** *Under the assumptions of Lemma B.5, fix a set $\mathbf{T} \subseteq \mathbf{A} \cup \mathbf{X}$. Then, for any intervention $\mathbf{A} \leftarrow \mathbf{a}'$ and observational values $\mathbf{a}, \mathbf{x}$ with $\mathbb{P}_{\mathbf{X},\mathbf{A}}(\mathbf{x}, \mathbf{a}) \neq 0$, there exist a point $\mathbf{t}$ in the support of $\mathbf{T}$ such that $\mathbb{P}_{\mathbf{T}_{\mathbf{a}'}^*|\mathbf{A}=\mathbf{a},\mathbf{X}=\mathbf{x}}(\mathbf{t}) = \mathbb{1}_\mathbf{t}$.*

*Proof.* In this proof, we use the notation introduced in Lemma B.5. Consider a set of nodes $\mathbf{T} \subseteq \mathbf{A} \cup \mathbf{X}$, and let $\mathbf{a}$ be a point in the support of $\mathbf{A}$. We denote with $\mathbf{a_T}$ the restriction of $\mathbf{a}$ to $\mathbf{A} \cap \mathbf{T}$. We prove the claim with a induction argument. To this end, we define $\mathsf{depth}(\mathbf{T})$ as the length of the longest directed path $\pi$ in $\mathcal{G}$ from any node in $\mathbf{A} \cup \mathbf{U_T}$ to any node in $\mathbf{T}$ (see Figure 6).[11]

(Base case). We first assume that with $\mathsf{depth}(\mathbf{T}) = 0$. We further assume w.l.o.g. that $\mathbf{T} \cap \mathbf{A} = \emptyset$, since the variables $\mathbf{T} \cap \mathbf{A}$ are fixed by the intervention $\mathbf{A} \leftarrow \mathbf{a}'$. Denote with $\mathbf{T} = g_\mathbf{T}(\mathsf{pa}(\mathbf{T}), \mathbf{U_T})$ the structural equation for the joint random variable $\mathbf{T}$. Since $\mathsf{depth}(\mathbf{T}) = 0$, it holds $\mathsf{pa}(\mathbf{T}) \subseteq \mathbf{A}$.[12] Then, the random variable $\mathbf{T}_{\mathbf{a}'}^*$ is defined by the formula $\mathbf{T}_{\mathbf{a}'}^* = g_\mathbf{T}(\mathbf{a}'\,|_\mathbf{T}, \mathbf{U_T})$. By Lemma B.5, there exist a value $\mathbf{u}'$ in the support of $\mathbf{U_T}$ such that $\mathbb{P}_{\mathbf{U_T}|\mathbf{A}=\mathbf{a},\mathbf{X}=\mathbf{x}}(\mathbf{u}) = \mathbb{1}_{\mathbf{u}'}$. It follows that $\mathbb{P}_{\mathbf{T}_{\mathbf{a}'}^*|\mathbf{A}=\mathbf{a},\mathbf{X}=\mathbf{x}}(\mathbf{t}) = \mathbb{1}_{\mathbf{t}'}$ with $\mathbf{t} := g_\mathbf{T}(\mathbf{a}'\,|_\mathbf{T}, \mathbf{u}')$.

(Inductive step). We assume by induction that the claim holds for all $\mathbf{T}$ such that $\mathsf{depth}(\mathbf{T}) \leq k$, and we prove the claim for a set $\mathbf{T}$ of depth $\mathsf{depth}(\mathbf{T}) = k + 1$. Since $\mathsf{depth}(\mathsf{pa}(\mathbf{T})) < \mathsf{depth}(\mathbf{T}) = k + 1$, by the inductive hypothesis, it holds

$$\mathbb{P}_{\mathsf{pa}(\mathbf{T})_{a'}^*|\mathbf{A}=\mathbf{a},\mathbf{X}=\mathbf{x}}(\mathbf{x}) = \mathbb{1}_{\mathbf{t}'}, \tag{4}$$

for a point $\mathbf{t}'$ in the support of $\mathsf{pa}(\mathbf{T})$. Again, consider the structural equation $\mathbf{T} = g_\mathbf{T}(\mathsf{pa}(\mathbf{T}), \mathbf{U_T})$ for the joint random variable $\mathbf{T}$. In this case, the random variable $\mathbf{T}_{\mathbf{a}'}^*$ is defined by the formula $\mathbf{T}_{\mathbf{a}'}^* = g_\mathbf{T}(\mathsf{pa}(\mathbf{T})_{a'}^*, \mathbf{U_T})$. By Lemma B.5, there exist a value $\mathbf{u}'$ in the support of $\mathbf{U_T}$ such that $\mathbb{P}_{\mathbf{U_T}|\mathbf{A}=\mathbf{a},\mathbf{X}=\mathbf{x}}(\mathbf{u}) = \mathbb{1}_{\mathbf{u}'}$. Combining this observation with Eq. (4), we get $\mathbb{P}_{\mathbf{T}_{a'}^*|\mathbf{A}=\mathbf{a},\mathbf{X}=\mathbf{x}}(\mathbf{t}) = \mathbb{1}_{g_\mathbf{T}(\mathbf{t}',\mathbf{u}')}$. The claim follows by defining $\mathbf{t} := g_\mathbf{T}(\mathbf{t}', \mathbf{u}')$. □

We use the results above to prove a third auxiliary lemma.
**Lemma B.7.** *Under the assumptions of Lemma B.5 and Lemma B.6, for any intervention $\mathbf{A} \leftarrow \mathbf{a}'$ and observational values $\mathbf{a}, \mathbf{x}$, there exist an intervention $\mathbf{X} \leftarrow \mathbf{x}'$ such that $\mathbb{P}_{\mathbf{Y}_{\mathbf{a}',\mathbf{x}'}|\mathbf{A}=\mathbf{a},\mathbf{X}=\mathbf{x}}(\mathbf{y}) = \mathbb{P}_{\mathbf{Y}_{\mathbf{a}'}|\mathbf{A}=\mathbf{a},\mathbf{X}=\mathbf{x}}(\mathbf{y})$.*

*Proof.* First note that by lemma B.6 there exists a point $\mathbf{x}'$ such that $\mathbb{P}_{\mathbf{X}_{\mathbf{a}'}^*|\mathbf{A}=\mathbf{a},\mathbf{X}=\mathbf{x}}(\mathbf{y}) = \mathbb{1}_{\mathbf{x}'}$. We choose this point to define the intervention $\mathbf{X} \leftarrow \mathbf{x}'$ as in the statement of this lemma. Denote with $\mathcal{G}_{\mathbf{a}'}$ the causal graph after performing the intervention $\mathbf{A} \leftarrow \mathbf{a}'$. Furthermore, denote with $\mathcal{G}_{\mathbf{a}',\mathbf{x}'}$ the causal graph after performing the interventions $\mathbf{A} \leftarrow \mathbf{a}'$ and $\mathbf{X} \leftarrow \mathbf{x}'$. We first prove that for any node $\mathbf{V}$ of $\mathcal{G}_{\mathbf{a}'}$ and $\mathcal{G}_{\mathbf{a}',\mathbf{x}'}$ it holds

$$\mathbb{P}_{\mathbf{V}|\mathbf{A}=\mathbf{a},\mathbf{X}=\mathbf{x}}(\mathbf{v}) = \mathbb{P}_{\mathbf{V}|\mathbf{A}=\mathbf{a},\mathbf{X}=\mathbf{x},\mathbf{X}_{\mathbf{a}'}^*=\mathbf{x}'}(\mathbf{v}). \tag{5}$$

In fact, it holds

$$\mathbb{P}_{\mathbf{V}|\mathbf{A}=\mathbf{a},\mathbf{X}=\mathbf{x}}(\mathbf{v}) = \int \mathbb{P}_{\mathbf{V}|\mathbf{A}=\mathbf{a},\mathbf{X}=\mathbf{x},\mathbf{X}_{\mathbf{a}'}^*=\mathbf{x}'}(\mathbf{v}) d\mathbb{P}_{\mathbf{X}_{\mathbf{a}'}^*|\mathbf{A}=\mathbf{a},\mathbf{X}=\mathbf{x}}(\mathbf{x}') \qquad \text{(by marginalization)}$$

$$= \int \mathbb{P}_{\mathbf{V}|\mathbf{A}=\mathbf{a},\mathbf{X}=\mathbf{x},\mathbf{X}_{\mathbf{a}'}^*=\mathbf{x}'}(\mathbf{v}) d\mathbb{1}_{\mathbf{x}'} \qquad \text{(by lemma B.6 with } \mathbf{T} = \mathbf{X})$$

$$= \mathbb{P}_{\mathbf{V}|\mathbf{A}=\mathbf{a},\mathbf{X}=\mathbf{x},\mathbf{X}_{\mathbf{a}'}^*=\mathbf{x}'}(\mathbf{v}). \qquad \text{(by marginalization)}$$

---

[11] Note that since $\mathsf{pa}(X) \in \mathbf{X} \cup \mathbf{A}$ for all $X \in \mathbf{T}$, then any such path $\pi$ only consists of nodes that are in $\mathbf{X} \cup \mathbf{A} \cup \mathbf{U_T}$.
[12] Here, $\mathsf{pa}(\mathbf{T})$ can also be the empty set. In this case, $\mathbf{T} = \mathbf{U_T}$ is an exogenous random variable and the proof follows.

Hence, Eq. (5) holds. It follows that

$$
\begin{aligned}
\mathbb{P}_{\mathbf{Y}^*_{\mathbf{a}',\mathbf{x}'}|\mathbf{A}=\mathbf{a},\mathbf{X}=\mathbf{x}}(\mathbf{y}) &= \mathbb{P}_{\mathbf{Y}^*_{\mathbf{a}',\mathbf{x}'}|\mathbf{A}=\mathbf{a},\mathbf{X}=\mathbf{x},\mathbf{X}^*_{\mathbf{a}'}=\mathbf{x}'}(\mathbf{y}) && \text{(Eq. (5) with } \mathbf{V}=\mathbf{Y}^*_{\mathbf{a}',\mathbf{x}'}) \\
&= \mathbb{P}_{\mathbf{Y}^*_{\mathbf{a}'}|\mathbf{A}=\mathbf{a},\mathbf{X}=\mathbf{x},\mathbf{X}^*_{\mathbf{a}'}=\mathbf{x}'}(\mathbf{y}) && \text{(Axiom of Consistency (Pearl, 2000))} \\
&= \mathbb{P}_{\mathbf{Y}^*_{\mathbf{a}'}|\mathbf{A}=\mathbf{a},\mathbf{X}=\mathbf{x}}(\mathbf{y}). && \text{(Eq. (5) with } \mathbf{V}=\mathbf{Y}^*_{\mathbf{a}'})
\end{aligned}
$$

Hence, the claim holds. □

We now prove our main result.

*Proof of Theorem 3.2.* Following the notation of Theorem B.2, define the set $\mathbf{X} = \mathbf{W} \setminus \mathbf{A}$. Note that, using this notation, the assumption $\mathbf{Y} \perp\!\!\!\perp \mathbf{A}, \mathbf{W} \mid \mathbf{S}$ can be written as $\mathbf{Y} \perp\!\!\!\perp \mathbf{A}, \mathbf{X} \mid \mathbf{S}$. Suppose that it holds

$$
\mathbb{P}_{\mathbf{Y}^*_{\mathbf{a}',\mathbf{x}'}|\mathbf{A}=\mathbf{a},\mathbf{X}=\mathbf{x}}(\mathbf{y}) = \int \mathbb{P}_{\mathbf{Y}|\mathbf{A}=\mathbf{a}',\mathbf{X}=\mathbf{x}',\mathbf{S}=\mathbf{s}}(\mathbf{y}) d\mathbb{P}_{\mathbf{S}|\mathbf{A}=\mathbf{a},\mathbf{X}=\mathbf{x}}(\mathbf{s}) \tag{6}
$$

for any intervention $\mathbf{A}, \mathbf{X} \leftarrow \mathbf{a}', \mathbf{x}'$, and for any possible value $\mathbf{w}$ attained by $\mathbf{W}$. Assuming that Eq. (6) holds, we have that

$$
\begin{aligned}
\mathbb{P}_{\mathbf{Y}^*_{\mathbf{a}',\mathbf{x}'}|\mathbf{A}=\mathbf{a},\mathbf{X}=\mathbf{x}}(\mathbf{y}) &= \int \mathbb{P}_{\mathbf{Y}|\mathbf{A}=\mathbf{a}',\mathbf{X}=\mathbf{x}',\mathbf{S}=\mathbf{s}}(\mathbf{y}) d\mathbb{P}_{\mathbf{S}|\mathbf{A}=\mathbf{a},\mathbf{X}=\mathbf{x}}(\mathbf{s}) && \text{(assuming Eq. (6))} \\
&= \int \mathbb{P}_{\mathbf{Y}|\mathbf{A}=\mathbf{a},\mathbf{X}=\mathbf{x}'',\mathbf{S}=\mathbf{s}}(\mathbf{y}) d\mathbb{P}_{\mathbf{S}|\mathbf{A}=\mathbf{a},\mathbf{X}=\mathbf{x}}(\mathbf{s}) && (\mathbf{Y} \perp\!\!\!\perp \mathbf{A}, \mathbf{X} \mid \mathbf{S}) \\
&= \mathbb{P}_{\mathbf{Y}^*_{\mathbf{a},\mathbf{x}''}|\mathbf{A}=\mathbf{a},\mathbf{X}=\mathbf{x}}(\mathbf{y}), && \text{(assuming Eq. (6))} \tag{7}
\end{aligned}
$$

for any value $\mathbf{x}''$ in the support of $\mathbf{X}$. To conclude, define the set $\mathbf{T} = \mathbf{A} \setminus \mathbf{W}$. Note that $\mathbf{W} \cup \mathbf{T} = \mathbf{A} \cup \mathbf{X}$. It follows that

$$
\begin{aligned}
\mathbb{P}_{\mathbf{Y}^*_{\mathbf{a}'}|\mathbf{W}=\mathbf{w}}(\mathbf{y}) &= \int \mathbb{P}_{\mathbf{Y}^*_{\mathbf{a}'}|\mathbf{A}=\mathbf{a},\mathbf{X}=\mathbf{x}}(\mathbf{y}) d\mathbb{P}_{\mathbf{T}|\mathbf{W}=\mathbf{w}}(\mathbf{t}) && \text{(by conditioning)} \\
&= \int \mathbb{P}_{\mathbf{Y}^*_{\mathbf{a}',\mathbf{x}'}|\mathbf{A}=\mathbf{a},\mathbf{X}=\mathbf{x}}(\mathbf{y}) d\mathbb{P}_{\mathbf{T}|\mathbf{W}=\mathbf{w}}(\mathbf{t}) && \text{(by Lemma B.7)} \\
&= \int \mathbb{P}_{\mathbf{Y}^*_{\mathbf{a},\mathbf{x}''}|\mathbf{A}=\mathbf{a},\mathbf{X}=\mathbf{x}}(\mathbf{y}) d\mathbb{P}_{\mathbf{T}|\mathbf{W}=\mathbf{w}}(\mathbf{t}) && \text{(by Eq. (7))} \\
&= \int \mathbb{P}_{\mathbf{Y}^*_{\mathbf{a}}|\mathbf{A}=\mathbf{a},\mathbf{X}=\mathbf{x}}(\mathbf{y}) d\mathbb{P}_{\mathbf{T}|\mathbf{W}=\mathbf{w}}(\mathbf{t}) && \text{(by Lemma B.7)} \\
&= \mathbb{P}_{\mathbf{Y}^*_{\mathbf{a}}|\mathbf{W}=\mathbf{w}}(\mathbf{y}). && \text{(by unconditioning)} \tag{8}
\end{aligned}
$$

Note that the claim follows follows from Eq. (8). The proof of Theorem 3.2 thus boils down to proving Eq. (6).

$$
\begin{aligned}
&\mathbb{P}_{\mathbf{Y}^*_{\mathbf{a}',\mathbf{x}'}|\mathbf{A}=\mathbf{a},\mathbf{X}=\mathbf{x}}(\mathbf{y}) \\
&= \int \mathbb{P}_{\mathbf{Y}^*_{\mathbf{a}',\mathbf{x}'}|\mathbf{A}=\mathbf{a},\mathbf{X}=\mathbf{x},\mathbf{S}=\mathbf{s}}(\mathbf{y}) d\mathbb{P}_{\mathbf{S}|\mathbf{A}=\mathbf{a},\mathbf{X}=\mathbf{x}}(\mathbf{s}) && \text{(by conditioning)} \\
&= \int \mathbb{P}_{\mathbf{Y}^*_{\mathbf{a}',\mathbf{x}'}|\mathbf{A}=\mathbf{a}',\mathbf{X}=\mathbf{x}',\mathbf{S}=\mathbf{s}}(\mathbf{y}) d\mathbb{P}_{\mathbf{S}|\mathbf{A}=\mathbf{a},\mathbf{X}=\mathbf{x}}(\mathbf{s}) && \text{(by Theorem B.2)} \\
&= \int \mathbb{P}_{\mathbf{Y}^*|\mathbf{A}=\mathbf{a}',\mathbf{X}=\mathbf{x}',\mathbf{S}=\mathbf{s}}(\mathbf{y}) d\mathbb{P}_{\mathbf{S}|\mathbf{A}=\mathbf{a},\mathbf{X}=\mathbf{x}}(\mathbf{s}) && \text{(Axiom of Consistency (Pearl, 2000))}
\end{aligned}
$$

$$\tag{9}$$

and Eq. (6) follows. □

## C  Proof of Theorem 3.4

We prove that the HSCIC can be used to promote conditional independence, using a similar technique as Park & Muandet (2020). The following theorem holds.

**Theorem 3.4** (Theorem 5.4 by Park & Muandet (2020)). *If the kernel $k$ of $\mathcal{H}_{\mathbf{X}} \otimes \mathcal{H}_{\mathbf{A} \cup \mathbf{W}}$ is characteristic[13], $HSCIC(\mathbf{Y}, \mathbf{A} \cup \mathbf{W} \mid \mathbf{S}) = 0$ almost surely if and only if $\mathbf{Y} \perp\!\!\!\perp \mathbf{A} \cup \mathbf{W} \mid \mathbf{S}$.*

*Proof.* By definition, we can write $HSCIC(\mathbf{Y}, \mathbf{A} \cup \mathbf{W} \mid \mathbf{S}) = H_{\mathbf{Y}, \mathbf{A} \cup \mathbf{W} \mid \mathbf{S}} \circ \mathbf{S}$, where $H_{\mathbf{Y}, \mathbf{A} \cup \mathbf{W} \mid \mathbf{S}}$ is a real-valued deterministic function. Hence, the HSCIC is a real-valued random variable, defined over the same domain $\Omega_{\mathbf{S}}$ of the random variable $\mathbf{X}$.

We first prove that if $HSCIC(\mathbf{Y}, \mathbf{A} \cup \mathbf{W} \mid \mathbf{S}) = 0$ almost surely, then it holds $\mathbf{Y} \perp\!\!\!\perp \mathbf{A} \cup \mathbf{W} \mid \mathbf{S}$. To this end, consider an event $\Omega' \subseteq \Omega_{\mathbf{X}}$ that occurs almost surely, and such that it holds $(H_{\mathbf{Y}, \mathbf{A} \cup \mathbf{W} \mid \mathbf{X}} \circ \mathbf{X})(\omega) = 0$ for all $\omega \in \Omega'$. Fix a sample $\omega \in \Omega'$, and consider the corresponding value $\mathbf{s}_{\omega} = \mathbf{S}(\omega)$, in the support of $\mathbf{S}$. It holds

$$\int k(\mathbf{y} \otimes [\mathbf{a}, \mathbf{w}], \, \cdot \,) d\mathbb{P}_{\mathbf{Y}, \mathbf{A} \cup \mathbf{W} \mid \mathbf{S} = \mathbf{s}_{\omega}} = \mu_{\mathbf{Y}, \mathbf{A} \cup \mathbf{W} \mid \mathbf{S} = \mathbf{s}_{\omega}} \qquad \text{(by definition)}$$

$$= \mu_{\mathbf{Y} \mid \mathbf{S} = \mathbf{s}_{\omega}} \otimes \mu_{\mathbf{A} \cup \mathbf{W} \mid \mathbf{S} = \mathbf{s}_{\omega}} \qquad \text{(since } \omega \in \Omega')$$

$$= \int k_{\mathbf{Y}}(\mathbf{y}, \, \cdot \,) d\mathbb{P}_{\mathbf{Y} \mid \mathbf{S} = \mathbf{s}_{\omega}} \otimes \int k_{\mathbf{A} \cup \mathbf{W}}([\mathbf{a}, \mathbf{w}], \, \cdot \,) d\mathbb{P}_{\mathbf{A} \cup \mathbf{W} \mid \mathbf{S} = \mathbf{s}_{\omega}} \qquad \text{(by definition )}$$

$$= \int k_{\mathbf{Y}}(\mathbf{y}, \, \cdot \,) \otimes k_{\mathbf{A} \cup \mathbf{W}}([\mathbf{a}, \mathbf{w}], \, \cdot \,) d\mathbb{P}_{\mathbf{Y} \mid \mathbf{S} = \mathbf{s}_{\omega}} \mathbb{P}_{\mathbf{A} \cup \mathbf{W} \cup \mathbf{W} \mid \mathbf{S} = \mathbf{s}_{\omega}}, \qquad \text{(by Fubini's Theorem)}$$

with $k_{\mathbf{Y}}$ and $k_{\mathbf{A} \cup \mathbf{W}}$ the kernels of $\mathcal{H}_{\mathbf{Y}}$ and $\mathcal{H}_{\mathbf{A} \cup \mathbf{W}}$ respectively. Since the kernel $k$ of the tensor product space $\mathcal{H}_{\mathbf{Y}} \otimes \mathcal{H}_{\mathbf{A} \cup \mathbf{W}}$ is characteristic, then the kernels $k_{\mathbf{Y}}$ and $k_{\mathbf{A} \cup \mathbf{W}}$ are also characteristic. Hence, it holds $\mathbb{P}_{\mathbf{Y}, \mathbf{A} \mid \mathbf{S} = \mathbf{s}_{\omega}} = \mathbb{P}_{\mathbf{Y} \mid \mathbf{S} = \mathbf{s}_{\omega}} \mathbb{P}_{\mathbf{A} \mid \mathbf{S} = \mathbf{s}_{\omega}}$ for all $\omega \in \Omega'$. Since the event $\Omega'$ occurs almost surely, then $\mathbb{P}_{\mathbf{Y}, \mathbf{A} \mid \mathbf{S} = \mathbf{s}_{\omega}} = \mathbb{P}_{\mathbf{Y} \mid \mathbf{S} = \mathbf{s}_{\omega}} \mathbb{P}_{\mathbf{A} \mid \mathbf{S} = \mathbf{s}_{\omega}}$ almost surely, that is $\mathbf{Y} \perp\!\!\!\perp \mathbf{A} \cup \mathbf{W} \mid \mathbf{S}$.

Assume now that $\mathbf{Y} \perp\!\!\!\perp \mathbf{A} \cup \mathbf{W} \mid \mathbf{S}$. By definition there exists an event $\Omega'' \subseteq \Omega_{\mathbf{S}}$ such that $\mathbb{P}_{\mathbf{Y}, \mathbf{A} \cup \mathbf{W} \mid \mathbf{S} = \mathbf{s}_{\omega}} = \mathbb{P}_{\mathbf{Y} \mid \mathbf{S} = \mathbf{s}_{\omega}} \mathbb{P}_{\mathbf{A} \cup \mathbf{W} \mid \mathbf{S} = \mathbf{s}_{\omega}}$ for all samples $\omega \in \Omega''$, with $\mathbf{s}_{\omega} = \mathbf{S}(\omega)$. It holds

$$\mu_{\mathbf{Y}, \mathbf{A} \cup \mathbf{W} \mid \mathbf{S} = \mathbf{s}_{\omega}} = \int k(\mathbf{y} \otimes [\mathbf{a}, \mathbf{w}], \, \cdot \,) d\mathbb{P}_{\mathbf{Y}, \mathbf{A} \cup \mathbf{W} \mid \mathbf{S} = \mathbf{s}_{\omega}} \qquad \text{(by definition)}$$

$$= \int k(\mathbf{y} \otimes [\mathbf{a}, \mathbf{w}], \, \cdot \,) d\mathbb{P}_{\mathbf{Y} \mid \mathbf{S} = \mathbf{s}_{\omega}} \mathbb{P}_{\mathbf{A} \cup \mathbf{W} \mid \mathbf{S} = \mathbf{s}_{\omega}} \qquad \text{(since } \omega \in \Omega')$$

$$= \int k_{\mathbf{Y}}(\mathbf{y}, \, \cdot \,) k_{\mathbf{A} \cup \mathbf{W}}([\mathbf{a}, \mathbf{w}], \, \cdot \,) d\mathbb{P}_{\mathbf{Y} \mid \mathbf{S} = \mathbf{s}_{\omega}} \mathbb{P}_{\mathbf{A} \cup \mathbf{W} \mid \mathbf{S} = \mathbf{s}_{\omega}} \qquad \text{(by definition of } k)$$

$$= \int k_{\mathbf{Y}}(\mathbf{y}, \, \cdot \,) d\mathbb{P}_{\mathbf{Y} \mid \mathbf{S} = \mathbf{s}_{\omega}} \otimes \int k_{\mathbf{A} \cup \mathbf{W}}([\mathbf{a}, \mathbf{w}], \, \cdot \,) d\mathbb{P}_{\mathbf{A} \cup \mathbf{W} \mid \mathbf{S} = \mathbf{s}_{\omega}} \qquad \text{(by Fubini's Theorem)}$$

$$= \mu_{\mathbf{Y} \mid \mathbf{S} = \mathbf{s}_{\omega}} \otimes \mu_{\mathbf{A} \cup \mathbf{W} \mid \mathbf{S} = \mathbf{s}_{\omega}}. \qquad \text{(by definition)}$$

The claim follows. $\square$

## D  Conditional kernel mean embeddings and the HSCIC

The notion of conditional kernel mean embeddings has already been studied in the literature. We show that, under stronger assumptions, our definition is equivalent to the definition by Park & Muandet (2020). In this section, without loss of generality we will assume that $\mathbf{W} = \emptyset$ and we will refer to the conditioning set as $\mathbf{Z}$.

### D.1  Conditional kernel mean embeddings and conditional independence

We show that, under stronger assumptions, the HSCIC can be defined using the Bochner conditional expected value. The Bochner conditional expected value is defined as follows.

---

[13]The tensor product kernel $k$ is characteristic if $\mathbb{P}_{\mathbf{Y}, \mathbf{A} \cup \mathbf{W}} \mapsto \mathbb{E}_{\mathbf{y}, [\mathbf{a}, \mathbf{w}]}[k(\, \cdot \,, \mathbf{y} \otimes [\mathbf{a}, \mathbf{w}])]$ is injective.

**Definition D.1.** *Fix two random variables* $\mathbf{Y}$, $\mathbf{Z}$ *taking value in a Banach space* $\mathcal{H}$, *and denote with* $(\Omega, \mathcal{F}, \mathbb{P})$ *their joint probability space. Then, the Bochner conditional expectation of* $\mathbf{Y}$ *given* $\mathbf{Z}$ *is any* $\mathcal{H}$-*valued random variable* $\mathbf{X}$ *such that*

$$\int_E \mathbf{Y}d\mathbb{P} = \int_E \mathbf{X}d\mathbb{P}$$

*for all* $E \in \sigma(\mathbf{Z}) \subseteq \mathcal{F}$, *with* $\sigma(\mathbf{Z})$ *the* $\sigma$-*algebra generated by* $\mathbf{Z}$. *We denote with* $\mathbb{E}\left[\mathbf{Y} \mid \mathbf{Z}\right]$ *the Bochner expected value. Any random variable* $\mathbf{X}$ *as above is a version of* $\mathbb{E}\left[\mathbf{Y} \mid \mathbf{Z}\right]$.

The existence and almost sure uniqueness of the conditional expectation are shown in Dinculeanu (2000). Given a RKHS $\mathcal{H}$ with kernel $k$ over the support of $\mathbf{Y}$, Park & Muandet (2020) define the corresponding conditional kernel mean embedding as

$$\mu_{\mathbf{Y}|\mathbf{Z}} := \mathbb{E}\left[k(\cdot, \mathbf{y}) \mid \mathbf{Z}\right].$$

Note that, according to this definition, $\mu_{\mathbf{Y}|\mathbf{Z}}$ is an $\mathcal{H}$-valued random variable, not a single point of $\mathcal{H}$. Park & Muandet (2020) use this notion to define the HSCIC as follows.

**Definition D.2** (The HSCIC according to Park & Muandet (2020)). *Consider (sets of) random variables* $\mathbf{Y}$, $\mathbf{A}$, $\mathbf{Z}$, *and consider two RKHS* $\mathcal{H}_{\mathbf{Y}}$, $\mathcal{H}_{\mathbf{A}}$ *over the support of* $\mathbf{Y}$ *and* $\mathbf{A}$ *respectively. The HSCIC between* $\mathbf{Y}$ *and* $\mathbf{A}$ *given* $\mathbf{Z}$ *is defined as the real-valued random variable*

$$\omega \mapsto \left\| \mu_{\mathbf{Y},\mathbf{A}|\mathbf{Z}}(\omega) - \mu_{\mathbf{Y}|\mathbf{Z}}(\omega) \otimes \mu_{\mathbf{A}|\mathbf{Z}}(\omega) \right\|,$$

*for all samples* $\omega$ *in the domain* $\Omega_{\mathbf{Z}}$ *of* $\mathbf{Z}$. *Here,* $\|\cdot\|$ *the metric induced by the inner product of the tensor product space* $\mathcal{H}_{\mathbf{Y}} \otimes \mathcal{H}_{\mathbf{Z}}$.

We show that, under more restrictive assumptions, Definition D.2 can be used to promote conditional independence. To this end, we use the notion of a regular version.

**Definition D.3** (Regular Version, following Definition 2.4 by Çinlar (2011)). *Consider two random variables* $\mathbf{Y}$, $\mathbf{Z}$, *and consider the induced measurable spaces* $(\Omega_{\mathbf{Y}}, \mathcal{F}_{\mathbf{Y}})$ *and* $(\Omega_{\mathbf{Z}}, \mathcal{F}_{\mathbf{Z}})$. *A regular version* $Q$ *for* $\mathbb{P}_{\mathbf{Y}|\mathbf{Z}}$ *is a mapping* $Q \colon \Omega_{\mathbf{Z}} \times \mathcal{F}_{\mathbf{Y}} \to [0, +\infty] \colon (\omega, \mathbf{y}) \mapsto Q_{\omega}(\mathbf{y})$ *such that: (i) the map* $\omega \mapsto Q_{\omega}(\mathbf{x})$ *is* $\mathcal{F}_{\mathbf{A}}$-*measurable for all* $\mathbf{y}$; *(ii) the map* $\mathbf{y} \mapsto Q_{\omega}(\mathbf{y})$ *is a measure on* $(\Omega_{\mathbf{Y}}, \mathcal{F}_{\mathbf{Y}})$ *for all* $\omega$; *(iii) the function* $Q_{\omega}(\mathbf{y})$ *is a version for* $\mathbb{E}\left[\mathbb{1}_{\{\mathbf{Y}=\mathbf{y}\}} \mid \mathbf{Z}\right]$.

The following theorem shows that the random variable as in Definition D.2 can be used to promote conditional independence.

**Theorem D.4** (Theorem 5.4 by Park & Muandet (2020)). *With the notation introduced above, suppose that the kernel* $k$ *of the tensor product space* $\mathcal{H}_{\mathbf{X}} \otimes \mathcal{H}_{\mathbf{A}}$ *is characteristic. Furthermore, suppose that* $\mathbb{P}_{\mathbf{Y},\mathbf{A}|\mathbf{X}}$ *admits a regular version. Then,* $\left\| \mu_{\mathbf{Y},\mathbf{A}|\mathbf{Z}}(\omega) - \mu_{\mathbf{Y}|\mathbf{Z}}(\omega) \otimes \mu_{\mathbf{A}|\mathbf{Z}}(\omega) \right\| = 0$ *almost surely if and only if* $\mathbf{Y} \perp\!\!\!\perp \mathbf{A} \mid \mathbf{Z}$.

Note that the assumption of the existence of a regular version is essential in Theorem D.4. In this work, HSCIC is not used for conditional independence testing but as a conditional independence measure.

## D.2 Equivalence with our approach

The following theorem shows that under the existence of a regular version, conditional kernel mean embeddings can be defined using the Bochner conditional expected value. To this end, we use the following theorem.

**Theorem D.5** (Following Proposition 2.5 by Çinlar (2011)). *Following the notation introduced in Definition D.3, suppose that* $\mathbb{P}_{\mathbf{Y}|\mathbf{Z}}(\cdot \mid \mathbf{Z})$ *admits a regular version* $Q_{\omega}(\mathbf{y})$. *Consider a kernel* $k$ *over the support of* $\mathbf{Y}$. *Then, the mapping*

$$\omega \mapsto \int k(\cdot, \mathbf{y}) dQ_{\omega}(\mathbf{y})$$

*is a version of* $\mathbb{E}\left[k(\cdot, \mathbf{y}) \mid \mathbf{Z}\right]$.

As a consequence of Theorem D.5, we prove the following result.

**Lemma D.6.** *Fix two random variables* $\mathbf{Y}$, $\mathbf{Z}$. *Suppose that* $\mathbb{P}_{\mathbf{Y}|\mathbf{Z}}$ *admits a regular version. Denote with* $\Omega_{\mathbf{Z}}$ *the domain of* $\mathbf{Z}$. *Then, there exists a subset* $\Omega \subseteq \Omega_{\mathbf{Z}}$ *that occurs almost surely, such that* $\mu_{\mathbf{Y}|\mathbf{Z}}(\omega) = \mu_{\mathbf{Y}|\mathbf{Z}=\mathbf{Z}(\omega)}$ *for all* $\omega \in \Omega$. *Here,* $\mu_{\mathbf{Y}|\mathbf{Z}=\mathbf{Z}(\omega)}$ *is the embedding of conditional measures as in Section 2.*

*Proof.* Let $Q_\omega(\mathbf{y})$ be a regular version of $\mathbb{P}_{\mathbf{Y}|\mathbf{Z}}$. Without loss of generality we may assume that it holds $\mathbb{P}_{\mathbf{Y}|\mathbf{Z}}(\mathbf{y} \mid \{\mathbf{Z} = \mathbf{Z}(\omega)\}) = Q_\omega(\mathbf{y})$. By Theorem D.5 there exists an event $\Omega \subseteq \Omega_{\mathbf{Z}}$ that occurs almost surely such that

$$\mu_{\mathbf{Y}|\mathbf{Z}}(\omega) = \mathbb{E}[k(\mathbf{y},\ \cdot\ ) \mid \mathbf{Z}](\omega) = \int k(\mathbf{y},\ \cdot\ )dQ_\omega(\mathbf{y}), \tag{10}$$

for all $\omega \in \Omega$. Then, for all $\omega \in \Omega$ it holds

$$\mu_{\mathbf{Y}|\mathbf{Z}}(\omega) = \int k(\mathbf{x},\ \cdot\ )dQ_\omega(\mathbf{x}) \qquad \text{(it follows from Eq. (10))}$$

$$= \int k(\mathbf{x},\ \cdot\ )d\mathbb{P}_{\mathbf{X}|\mathbf{A}}(\mathbf{x} \mid \{\mathbf{A} = \mathbf{A}(\omega)\}) \qquad (Q_\omega(\mathbf{y}) = \mathbb{P}_{\mathbf{Y}|\mathbf{Z}}(\mathbf{y} \mid \{\mathbf{Z} = \mathbf{Z}(\omega)\}))$$

$$= \mu_{\mathbf{X}|\{\mathbf{A}=\mathbf{A}(\omega)\}}, \qquad \text{(by definition as in Section 2)}$$

as claimed. $\qquad\qquad\square$

As a consequence of Lemma D.6, we can prove that the definition of the HSCIC by Park & Muandet (2020) is equivalent to ours. The following corollary holds.

**Corollary D.7.** *Consider (sets of) random variables $\mathbf{Y}$, $\mathbf{A}$, $\mathbf{Z}$, and consider two RKHS $\mathcal{H}_{\mathbf{Y}}$, $\mathcal{H}_{\mathbf{A}}$ over the support of $\mathbf{Y}$ and $\mathbf{A}$ respectively. Suppose that $\mathbb{P}_{\mathbf{Y},\mathbf{A}|\mathbf{Z}}(\cdot \mid \mathbf{Z})$ admits a regular version. Then, there exists a set $\Omega \subseteq \Omega_{\mathbf{A}}$ that occurs almost surely, such that*

$$\left\| \mu_{\mathbf{X},\mathbf{A}|\mathbf{Z}}(\omega) - \mu_{\mathbf{X}|\mathbf{Z}}(\omega) \otimes \mu_{\mathbf{A}|\mathbf{Z}}(\omega) \right\| = (H_{\mathbf{Y},\mathbf{A}|\mathbf{Z}} \circ \mathbf{Z})(\omega).$$

*Here, $H_{\mathbf{Y},\mathbf{A}|\mathbf{Z}}$ is a real-valued deterministic function, defined as*

$$H_{\mathbf{Y},\mathbf{A}|\mathbf{Z}}(\mathbf{z}) := \left\| \mu_{\mathbf{Y},\mathbf{A}|\mathbf{Z}=\mathbf{z}} - \mu_{\mathbf{Y}|\mathbf{Z}=\mathbf{z}} \otimes \mu_{\mathbf{A}|\mathbf{Z}=\mathbf{z}} \right\|,$$

*and $\|\cdot\|$ is the metric induced by the inner product of the tensor product space $\mathcal{H}_{\mathbf{X}} \otimes \mathcal{H}_{\mathbf{A}}$.*

We remark that the assumption of the existence of a regular version is essential in Corollary D.7.

# E  The cross-covariance operator

In this section, we show that under additional assumptions, our definition of conditional KMEs is equivalent to the definition based on the cross-covariance operator, under more restrictive assumptions. The definition of KMEs based on the cross-covariance operator requires the use of the following well-known result.

**Lemma E.1.** *Fix two RKHS $\mathcal{H}_{\mathbf{X}}$ and $\mathcal{H}_{\mathbf{Z}}$, and let $\{\varphi_i\}_{i=1}^{\infty}$ and $\{\psi_j\}_{j=1}^{\infty}$ be orthonormal bases of $\mathcal{H}_{\mathbf{X}}$ and $\mathcal{H}_{\mathbf{Z}}$ respectively. Denote with $\mathsf{HS}(\mathcal{H}_{\mathbf{X}}, \mathcal{H}_{\mathbf{Z}})$ the set of Hilbert-Schmidt operators between $\mathcal{H}_{\mathbf{X}}$ and $\mathcal{H}_{\mathbf{Z}}$. There is an isometric isomorphism between the tensor product space $\mathcal{H}_{\mathbf{X}} \otimes \mathcal{H}_{\mathbf{Z}}$ and $\mathsf{HS}(\mathcal{H}_{\mathbf{X}}, \mathcal{H}_{\mathbf{Z}})$, given by the map*

$$T \colon \sum_{i=1}^{\infty} \sum_{j=1}^{\infty} c_{i,j}\varphi_i \otimes \psi_j \mapsto \sum_{i=1}^{\infty} \sum_{j=1}^{\infty} c_{i,j}\langle\ \cdot\ , \varphi_i\rangle_{\mathcal{H}_{\mathbf{X}}} \psi_j.$$

For proof of this result see i.e., Park & Muandet (2020). This lemma allows us to define the cross-covariance operator between two random variables, using the operator $T$.

**Definition E.2** (Cross-Covariance Oprator). *Consider two random variables $\mathbf{X}$, $\mathbf{Z}$. Consider corresponding mean embeddings $\mu_{\mathbf{X},\mathbf{Z}}$, $\mu_{\mathbf{X}}$ and $\mu_{\mathbf{Z}}$, as defined in Section 3. The cross-covariance operator is defined as $\Sigma_{\mathbf{X},\mathbf{Z}} := T(\mu_{\mathbf{X},\mathbf{Z}} - \mu_{\mathbf{X}} \otimes \mu_{\mathbf{Z}})$. Here, $T$ is the isometric isomorphism as in Lemma E.1.*

It is well-known that the cross-covariance operator can be decomposed into the covariance of the marginals and the correlation. That is, there exists a unique bounded operator $\Lambda_{\mathbf{Y},\mathbf{Z}}$ such that

$$\Sigma_{\mathbf{Y},\mathbf{Z}} = \Sigma_{\mathbf{Y},\mathbf{Y}}^{1/2} \circ \Lambda_{\mathbf{Y},\mathbf{Z}} \circ \Sigma_{\mathbf{Z},\mathbf{Z}}^{1/2}$$

Using this notation, we define the *normalized conditional cross-covariance operator*. Given three random variables $\mathbf{Y}$, $\mathbf{A}$, $\mathbf{Z}$ and corresponding kernel mean embeddings, this operator is defined as

$$\Lambda_{\mathbf{Y},\mathbf{A}|\mathbf{Z}} := \Lambda_{\mathbf{Y},\mathbf{A}} - \Lambda_{\mathbf{Y},\mathbf{Z}} \circ \Lambda_{\mathbf{Z},\mathbf{A}}. \tag{11}$$

This operator was introduced by Fukumizu et al. (2007). The normalized conditional cross-covariance can be used to promote statistical independence, as shown in the following theorem.

**Theorem E.3** (Theorem 3 by Fukumizu et al. (2007)). *Following the notation introduced above, define the random variable* $\ddot{\mathbf{A}} \coloneqq (\mathbf{A}, \mathbf{Z})$. *Let* $\mathbb{P}_{\mathbf{Z}}$ *be the distribution of the random variable* $\mathbf{Z}$, *and denote with* $L^2(\mathbb{P}_{\mathbf{Z}})$ *the space of the square integrable functions with probability* $\mathbb{P}_{\mathbf{Z}}$. *Suppose that the tensor product kernel* $k_{\mathbf{Y}} \otimes k_{\mathbf{A}} \otimes k_{\mathbf{Z}}$ *is characteristic. Furthermore, suppose that* $\mathcal{H}_{\mathbf{Z}} + \mathbb{R}$ *is dense in* $L^2(\mathbb{P}_{\mathbf{Z}})$. *Then, it holds*

$$\Lambda_{\mathbf{Y},\ddot{\mathbf{A}}|\mathbf{Z}} = 0 \quad \text{if and only if} \quad \mathbf{Y} \perp\!\!\!\perp \mathbf{A} \mid \mathbf{Z}.$$

*Here,* $\Lambda_{\mathbf{Y},\ddot{\mathbf{A}}|\mathbf{Z}}$ *is an operator defined as in Eq.* (11).

By Theorem E.3, the operator $\Lambda_{\mathbf{Y},\ddot{\mathbf{A}}|\mathbf{Z}}$ can also be used to promote conditional independence. However, CIP is more straightforward since it requires less assumptions. In fact, Theorem E.3 requires to embed the variable $\mathbf{Z}$ in an RKHS. In contrast, CIP only requires the embedding of the variables $\mathbf{Y}$ and $\mathbf{A}$.

## F   Random fourier features

Random Fourier features is an approach to scaling up kernel methods for shift-invariant kernels (Rahimi & Recht, 2007). Recall that a shift-invariant kernel is a kernel of the form $k(\mathbf{z}, \mathbf{z}') = h_k(\mathbf{z} - \mathbf{z}')$, with $h_k$ a positive definite function.

Fourier features are defined via the following well-known theorem.

**Theorem F.1** (Bochner's Theorem). *For every shift-invariant kernel of the form* $k(\mathbf{z}, \mathbf{z}') = h_k(\mathbf{z} - \mathbf{z}')$ *with* $h_k(\mathbf{0}) = 1$, *there exists a probability probability density function* $\mathbb{P}_k(\eta)$ *such that*

$$k(\mathbf{z}, \mathbf{z}') = \int e^{-2\pi i \eta^T (\mathbf{z} - \mathbf{z}')} d\mathbb{P}_k.$$

Since both the kernel $k$ and the probability distribution $\mathbb{P}_k$ are real-valued functions, the integrand in Theorem F.1 ca be replaced by the function $\cos \eta^T(\mathbf{z} - \mathbf{z}')$, and we obtain the following formula

$$k(\mathbf{z}, \mathbf{z}') = \int \cos \eta^T(\mathbf{z} - \mathbf{z}') d\mathbb{P}_k = \mathbb{E}\left[\cos \eta^T(\mathbf{z} - \mathbf{z}')\right], \tag{12}$$

where the expected value is taken with respect to the distribution $\mathbb{P}_k(\eta)$. This equation allows to approximate the kernel $k(\mathbf{z}, \mathbf{z}')$, via the empirical mean of points $\eta_1, \ldots, \eta_l$ sampled independently according to $\mathbb{P}_k$. In fact, it is possible to prove exponentially fast convergence of an empirical estimate for $\mathbb{E}\left[\cos \eta^T(\mathbf{z} - \mathbf{z}')\right]$, as shown in the following theorem.

**Theorem F.2** (Uniform Convergence of Fourier Features, Claim 1 by Rahimi & Recht (2007)). *Following the notation introduced above, fix any compact subset* $\Omega$ *in the domain of* $k$, *and consider points* $\eta_1, \ldots, \eta_l$ *sampled independent according to the distribution* $\mathbb{P}_k$. *Define the function*

$$\hat{k}(\mathbf{z}, \mathbf{z}') \coloneqq \frac{1}{l} \sum_{j=1}^{l} \cos \eta_j^T(\mathbf{z} - \mathbf{z}'),$$

*for all* $(\mathbf{z}, \mathbf{z}') \in \Omega$. *Then, it holds*

$$\mathbb{P}\left(\sup_{\mathbf{z},\mathbf{z}'} \left|\hat{k}(\mathbf{z}, \mathbf{z}') - k(\mathbf{z}, \mathbf{z}')\right| \geq \varepsilon\right) \leq 2^8 \sigma_k \frac{\mathsf{diam}(\Omega)}{\varepsilon} \exp\left\{-\frac{\varepsilon^2 l}{4(d+1)}\right\}.$$

*Here* $\sigma_k^2$ *is the second moment of the Fourier transform of the kernel* $k$, *and* $d$ *is the dimension of the arrays* $\mathbf{z}$ *and* $\mathbf{z}'$.

By Theorem F.2, the estimated kernel $\hat{k}$ is a good approximation of the true kernel $k$ on the set $\Omega$.

Similarly, we can approximate the Kernel matrix using Random Fourier features. Following the notation introduced above, define the function

$$\zeta_{k,l}(\mathbf{z}) \coloneqq \frac{1}{\sqrt{l}}\left[\cos \eta_1^T \mathbf{z}, \ldots, \cos \eta_l^T \mathbf{z}\right] \tag{13}$$

with $\eta_1, \ldots, \eta_l$ sampled independent according to the distribution $\mathbb{P}_k$.

We can approximate the Kernel matrix using the functions defined as in Eq. (13). Consider $n$ samples $\mathbf{z}_1, \ldots, \mathbf{z}_n$, and denote with $Z$ the $n \times l$ matrix whose $i$-th row is given by $\zeta_{k,l}(\mathbf{z}_i)$. Similarly, denote with $Z^*$ the $l \times n$ matrix whose $i$-th column is given by $\zeta_{k,l}^*(\mathbf{z}_i)$. Then, we can approximate the kernel matrix as $\hat{K}_{\mathbf{Z}} \approx ZZ^*$.

We can also use this approximation to compute the kernel ridge regression parameters as in Section 3 using the formula $\hat{w}_{\mathbf{Y}|\mathbf{Z}}(\cdot) \approx (ZZ^* - n\lambda I)^{-1} \begin{bmatrix} k_{\mathbf{Z}}(\cdot, \mathbf{z}_1), \cdots, k_{\mathbf{Z}}(\cdot, \mathbf{z}_n) \end{bmatrix}^T$. Avron et al. (2017) argue that the approximate kernel ridge regression, as defined above, is an accurate estimate of the true distribution. Their argument is based on proving that the matrix $ZZ^* - n\lambda I$ is a *good approximation* of $\hat{K}_{\mathbf{Z}} - n\lambda I$. The notion of good approximation is clarified by the following definition.

**Definition F.3.** *Fix two Hermitian matrices $A$ and $B$ of the same size. We say that a matrix $A$ is a $\gamma$-spectral approximation of another matrix $B$, if it holds $(1 - \gamma)B \preceq A \preceq (1 + \gamma)B$. Here, the $\preceq$ symbol means that $A - (1 - \gamma)B$ is positive semi-definite, and that $(1 + \gamma)B - A$ is positive semi-definite.*

Avron et al. (2017) prove that $ZZ^* - n\lambda I$ is a $\gamma$- approximation of $\hat{K}_{\mathbf{Z}} - n\varepsilon I$, if the number of samples $\eta_1, \ldots, \eta_l$ is sufficiently large.

**Theorem F.4** (Theorem 7 by Avron et al. (2017)). *Fix a constant $\gamma \leq 1/2$. Consider $n$ samples $\mathbf{z}_1, \ldots, \mathbf{z}_n$, and denote with $\hat{K}_{\mathbf{Z}}$ the corresponding kernel matrix. Suppose that it holds $\|\hat{K}_{\mathbf{Z}}\|_2 \geq n\lambda$ for a constant $\lambda > 0$. Fix $\eta_1, \ldots, \eta_l$ samples with*

$$l \geq \frac{8}{3\gamma^2\lambda} \ln \frac{16 \, \mathsf{tr}_\lambda(\hat{K}_{\mathbf{Z}})}{\gamma}$$

*Then, the matrix $ZZ^* - n\lambda I$ is a $\gamma$- approximation of $\hat{K}_{\mathbf{Z}} - n\lambda I$ with probability at least $1 - \gamma$, for all $\gamma \in (0, 1)$. Here, $\mathsf{tr}_\lambda(\hat{K}_{\mathbf{Z}})$ is defined as the trace of the matrix $\hat{K}_{\mathbf{Z}}(\hat{K}_{\mathbf{Z}} + n\lambda I)^{-1}$.*

We conclude this section by illustrating the use of random Fourier features to approximate a simple Gaussian kernel. Suppose that we are given a kernel of the form

$$k(\mathbf{z}, \mathbf{z}') \coloneqq \exp\left\{ -\frac{1}{2}\sigma\|\mathbf{z} - \mathbf{z}'\|_2^2 \right\}.$$

Then, $k(\mathbf{z}, \mathbf{z}')$ can be estimated as in Theorem F.2, with $\eta_1, \ldots, \eta_l \sim \mathcal{N}(0, \Sigma)$, with $\Sigma \coloneqq \sigma^{-1}I$, with $I$ the identity matrix. The functions $\zeta_{k,l}(\mathbf{z})$ can be defined accordingly.

# G  Additional experiments and settings

This section contains detailed information on the experiments and additional results.

## G.1  Dataset for model performance with the use of the HSCIC

The data-generating mechanism corresponding to the results in Fig. 2 is the following:

$$\mathbf{Z} \sim \mathcal{N}(0, 1) \qquad \mathbf{A} = \mathbf{Z}^2 + \varepsilon_{\mathbf{A}}$$

$$\mathbf{L} = \exp\left\{ -\frac{1}{2}\mathbf{A}^2 \right\} \sin(2\mathbf{A}) + 2\mathbf{Z}\frac{1}{5}\varepsilon_{\mathbf{L}}$$

$$\mathbf{Y} = \frac{1}{2}\exp\{-\mathbf{L}\mathbf{Z}\} \cdot \sin(2\mathbf{L}\mathbf{Z}) + 5\mathbf{A} + \frac{1}{5}\varepsilon_{\mathbf{Y}},$$

where $\varepsilon_{\mathbf{A}} \sim \mathcal{N}(0, 1)$ and $\varepsilon_{\mathbf{Y}}, \varepsilon_{\mathbf{L}} \stackrel{i.i.d.}{\sim} \mathcal{N}(0, 0.1)$.

In the first experiment, Fig. 2 shows the results of feed-forward neural networks consisting of 8 hidden layers with 20 nodes each, connected with a rectified linear activation function (ReLU) and a linear final layer. Mini-batch size of 256 and the Adam optimizer with a learning rate of $10^{-3}$ for 1000 epochs were used.

Table 1: **Performance of the HSCIC against baselines** CF1 and CF2 on two synthetic datasets. Notably, in both scenarios it is possible to select $\gamma$ values for which CIP outperforms CF2 in **MSE** and **VCF** simultaneously.

| | Scenario 1 | | | Scenario 2 | | |
|---|---|---|---|---|---|---|
| | **MSE** $\times 10^6$ | **HSCIC** $\times 10^3$ | **VCF** $\times 10^3$ | **MSE** $\times 10^3$ | **HSCIC** $\times 10^2$ | **VCF** $\times 10^2$ |
| $\gamma = 0.001$ | $12 \pm 9$ | $45.38 \pm 0.41$ | $54.93 \pm 7.50$ | $0.0006$ | $35.64 \pm 0.32$ | $5.60 \pm 0.03$ |
| $\gamma = 0.01$ | $16 \pm 12$ | $45.35 \pm 0.41$ | $54.57 \pm 7.18$ | $0.0019$ | $35.44 \pm 0.33$ | $5.50 \pm 0.03$ |
| $\gamma = 0.1$ | $32 \pm 20$ | $45.11 \pm 0.43$ | $54.16 \pm 7.58$ | $0.11 \pm 0.006$ | $33.47 \pm 0.36$ | $4.46 \pm 0.04$ |
| $\gamma = 0.2$ | $81 \pm 14$ | $44.78 \pm 0.47$ | $53.59 \pm 7.90$ | $0.42 \pm 0.02$ | $31.38 \pm 0.38$ | $3.52 \pm 0.04$ |
| $\gamma = 0.3$ | $192 \pm 33$ | $43.92 \pm 0.52$ | $52.92 \pm 7.54$ | $0.82 \pm 0.04$ | $29.75 \pm 0.34$ | $2.50 \pm 0.04$ |
| $\gamma = 0.4$ | $384 \pm 58$ | $43.88 \pm 0.57$ | $52.06 \pm 7.25$ | $1.21 \pm 0.05$ | $28.63 \pm 0.33$ | $1.79 \pm 0.03$ |
| $\gamma = 0.5$ | $685 \pm 133$ | $43.26 \pm 0.65$ | $51.64 \pm 7.40$ | $1.56 \pm 0.08$ | $27.81 \pm 0.26$ | $1.1 \pm 0.01$ |
| $\gamma = 0.6$ | $1117 \pm 165$ | $42.47 \pm 0.73$ | $50.96 \pm 7.36$ | $1.84 \pm 0.11$ | $26.87 \pm 0.22$ | $0.79 \pm 0.01$ |
| $\gamma = 0.7$ | $1655 \pm 223$ | $42.11 \pm 0.80$ | $50.31 \pm 7.44$ | $2.11 \pm 0.14$ | $26.08 \pm 0.20$ | $0.49 \pm 0.01$ |
| $\gamma = 0.8$ | $2225 \pm 296$ | $41.87 \pm 0.84$ | $49.76 \pm 7.25$ | $2.37 \pm 0.15$ | $25.27 \pm 0.18$ | $0.31 \pm 0.01$ |
| $\gamma = 0.9$ | $2832 \pm 372$ | $41.52 \pm 0.92$ | $49.17 \pm 7.41$ | $2.58 \pm 0.17$ | $24.64 \pm 0.16$ | $0.21 \pm 0.01$ |
| $\gamma = 1.0$ | $3472 \pm 422$ | $38.37 \pm 0.97$ | $48.71 \pm 7.55$ | $2.77 \pm 0.19$ | $24.21 \pm 0.15$ | $0.14 \pm 0.01$ |
| CF1 | $10321 \pm 72$ | $41.37 \pm 0.58$ | $0 \pm 0.00$ | $4.59 \pm 0.4478$ | $25.01 \pm 0.25$ | $0 \pm 0.00$ |
| CF2 | $2728 \pm 272$ | $41.37 \pm 0.92$ | $59.50 \pm 10.35$ | $3.97 \pm 0.3479$ | $27.03 \pm 0.35$ | $2.62 \pm 0.81$ |

## G.2 Datasets and results for comparison with baselines

The comparison of our method CIP with the CF1 and CF2 is done on different simulated datasets. These will be referred to as Scenario 1 and Scenario 2. The data generating mechanism corresponding to the results in Fig. 2 (right) is the following:

$$\mathbf{Z} \sim \mathcal{N}(0,1) \qquad \mathbf{A} = \exp\left\{\frac{1}{2}\mathbf{Z}^2\right\} \cdot \sin(2\mathbf{Z}) + \varepsilon_{\mathbf{A}}$$

$$\mathbf{L} = (\mathbf{A} + 0.1\mathbf{Z}) \cdot \varepsilon_{\mathbf{L}}$$

$$\mathbf{Y} = \mathbf{A} + \mathbf{L} + 0.1 \cdot \sin(\mathbf{Z})$$

where $\varepsilon_{\mathbf{A}}, \varepsilon_{\mathbf{L}} \overset{i.i.d.}{\sim} \mathcal{N}(0,1)$ and $\varepsilon_{\mathbf{Y}} \overset{i.i.d.}{\sim} \mathcal{N}(0,0.1)$. This is referred to as Scenario 1. The data generating mechanism for Scenario 2 is the following:

$$\mathbf{Z} \sim \mathcal{N}(0,1) \qquad \mathbf{A} = \exp\left\{\frac{1}{2}\mathbf{Z}^2\right\} \cdot \sin(2\mathbf{Z}) + \varepsilon_{\mathbf{A}}$$

$$\mathbf{L} = \exp\left\{-\frac{1}{2}\mathbf{A}^2\right\} \cdot \varepsilon_{\mathbf{L}} + 2\mathbf{Z}$$

$$\mathbf{Y} = \frac{1}{2}\sin(\mathbf{ZL}) \cdot \exp\{-\mathbf{ZL}\} + \frac{1}{5}\varepsilon_{\mathbf{Y}},$$

where $\varepsilon_{\mathbf{A}}, \varepsilon_{\mathbf{L}} \overset{i.i.d.}{\sim} \mathcal{N}(0,1)$ and $\varepsilon_{\mathbf{Y}} \overset{i.i.d.}{\sim} \mathcal{N}(0,0.1)$. Fig. 2 (right) and Table 1 present the average and standard deviation resulting from 9 random seeds runs. For CIP, the same hyperparameters as in the previous setting are used. The MLPs implemented in CF1 and CF2 used for the prediction of $\hat{\mathbf{Y}}$ and the one used for the prediction of the $\mathbf{L}$ residuals in CF2 are all designed with similar architecture and training method. The MLP models consist of 8 hidden layers with 20 nodes each, connected with a rectified linear activation function (ReLU) and a linear final layer. During training, mini-batch size of 64 and the Adam optimizer with a learning rate of $10^{-3}$ for 200 epochs were used.

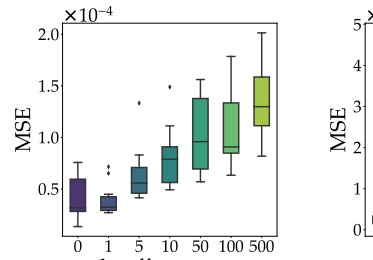 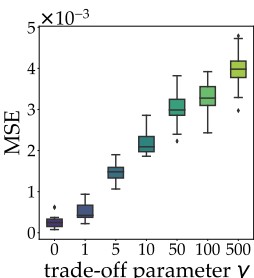 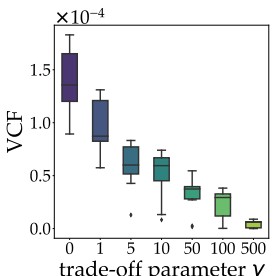 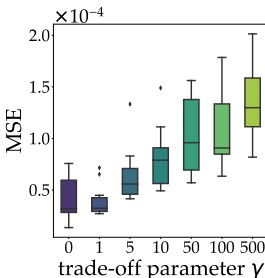

Figure 7: MSE , HSCIC, VCF for increasing dimension of $\mathbf{A}$ on synthetic data from Appendix G.3 with dimA = 20 (left) and dimA = 100 (right). All other variables are one-dimensional.

### G.3 Datasets and results for multi-dimensional variables experiments

The data-generating mechanisms for the multi-dimensional settings of Fig. 3 are now shown. Given dimA = $D_1 \geq 2$, the datasets were generated from:

$$\mathbf{Z} \sim \mathcal{N}(0,1) \qquad \mathbf{A}_i = \mathbf{Z}^2 + \varepsilon_{\mathbf{A}}^i \quad \text{for } i \in \{1, D_1\}$$

$$\mathbf{L} = \exp\left\{-\frac{1}{2}\mathbf{A}_1\right\} + \sum_{i=1}^{D_1} \mathbf{A}_i \cdot \sin(\mathbf{Z}) + 0.1 \cdot \varepsilon_{\mathbf{L}}$$

$$\mathbf{Y} = \exp\left\{-\frac{1}{2}\mathbf{A}_2\right\} \cdot \sum_{i=1}^{D_1} \mathbf{A}_i + \mathbf{LZ} + 0.1 \cdot \varepsilon_{\mathbf{Y}},$$

where $\varepsilon_{\mathbf{L}}, \varepsilon_{\mathbf{Y}} \overset{i.i.d}{\sim} \mathcal{N}(0, 0.1)$ and $\varepsilon_{\mathbf{A}}^1, ..., \varepsilon_{\mathbf{A}}^{D_1} \overset{i.i.d}{\sim} \mathcal{N}(0,1)$. In this experiment, the mini-batch size chosen is 512 and the same hyperparameters are used as in the previous settings. The neural network architecture is trained for 800 epochs. Fig. 7 present the results corresponding to 10 random seeds with different values of the trade-off parameter $\gamma$ corresponding to different values of dimA among $\{15, 100\}$. In all of the box plots, it is evident that there exists a trade-off between the accuracy and counterfactual invariance of the predictor. As the value of $\gamma$ increases, there is a consistent trend of augmenting counterfactual invariance (as evidenced by the decrease in the VCF metric). Similarly to the previous boxplots visualizations, the boxes represent the interquartile range (IQR), the horizontal line is the median, and whiskers show the minimum and maximum values, excluding the outliers (determined as a function of the inter-quartile range). Outliers are represented in the plot as dots.

### G.4 Image dataset

The simulation procedure for the results shown in Section 4.2 is the following.

$$\texttt{shape} \sim \mathbb{P}(\texttt{shape})$$
$$\texttt{y-pos} \sim \mathbb{P}(\texttt{y-pos})$$
$$\texttt{color} \sim \mathbb{P}(\texttt{color})$$
$$\texttt{orientation} \sim \mathbb{P}(\texttt{orientation})$$
$$\texttt{x-pos} = \text{round}(x), \quad \text{where } x \sim \mathcal{N}(\texttt{shape} + \texttt{y-pos}, 1)$$
$$\texttt{scale} = \text{round}\left(\left(\frac{\texttt{x-pos}}{24} + \frac{\texttt{y-pos}}{24}\right) \cdot \texttt{shape} + \epsilon_S\right)$$
$$\mathbf{Y} = e^{\texttt{shape}} \cdot \texttt{x-pos} + \texttt{scale}^2 \cdot \sin(\texttt{y-pos}) + \epsilon_Y,$$

where $\epsilon_S \sim \mathcal{N}(0,1)$ and $\epsilon_Y \sim \mathcal{N}(0, 0.01)$. The data has been generated via a matching procedure on the original dSprites dataset.

In Table 2, the hyperparameters of the layers of the convolutional neural network are presented. Each of the convolutional groups also has a ReLU activation function and a dropout layer. Two MLP architectures have

Table 2: Architecture of the convolutional neural network used for the image dataset, as described in Appendix G.4.

| layer | # filters | kernel size | stride size | padding size |
|---|---|---|---|---|
| convolution | 16 | 5 | 2 | 2 |
| max pooling | 1 | 3 | 2 | 0 |
| convolution | 64 | 5 | 1 | 2 |
| max pooling | 1 | 1 | 2 | 0 |
| convolution | 64 | 5 | 1 | 2 |
| max pooling | 1 | 2 | 1 | 0 |
| convolution | 16 | 5 | 1 | 3 |
| max pooling | 1 | 2 | 2 | 0 |

been used. The former takes as input the observed tabular features. It is composed by two hidden layers of 16 and 8 nodes respectively, connected with ReLU activation functions and dropout layers. The latter takes as input the concatenated outcomes of the CNN and the other MLP. It consists of three hidden layers of 8, 8 and 16 nodes, respectively.

### G.5 Fairness with continuous protected attributes

The pre-processing of the UCI Adult dataset was based upon the work of Chiappa & Pacchiano (2021). Referring to the causal graph in Fig. 8, a variational autoencoder (Kingma & Welling, 2014) was trained for each of the unobserved variables $\mathbf{H_m}$, $\mathbf{H_l}$ and $\mathbf{H_r}$. The prior distribution of these latent variables is assumed to be standard Gaussian. The posterior distributions $\mathbb{P}(\mathbf{H_m}|V)$, $\mathbb{P}(\mathbf{H_r}|V)$, $\mathbb{P}(\mathbf{H_d}|V)$ are modeled as 10-dimensional Gaussian distributions, whose means and variances are the outputs of the encoder.

The encoder architecture consists of a hidden layer of 20 hidden nodes with hyperbolic tangent activation functions, followed by a linear layer. The decoders have two linear layers with a hyperbolic tangent activation function. The training loss of the variational autoencoder consists of a reconstruction term (Mean-Squared Error for continuous variables and Cross-Entropy Loss for binary ones) and the Kullback–Leibler divergence between the posterior and the prior distribution of the latent variables. For training, we used the Adam optimizer with learning rate of $10^{-2}$, 100 epochs, mini-batch size 128.

The predictor $\hat{\mathbf{Y}}$ is the output of a feed-forward neural network consisting of a hidden layer with a hyperbolic tangent activation function and a linear final layer. In the training we used the Adam optimizer with learning rate $10^{-3}$, mini-batch size 128, and trained for 100 epochs. The choice of the network architecture is based on the work of Chiappa & Pacchiano (2021).

The estimation of counterfactual outcomes is based on a Monte Carlo approach. Given a data point, 500 values of the unobserved variables are sampled from the estimated posterior distribution. Given an interventional value for $A$, a counterfactual outcome is estimated for each of the sampled unobserved values. The final

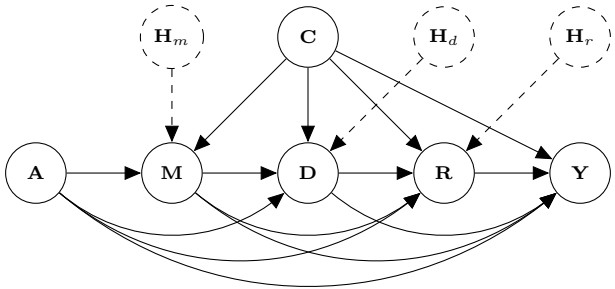

Figure 8: Assumed causal graph for the Adult dataset, as in Chiappa & Pacchiano (2021). The variables $\mathbf{H}_m$, $\mathbf{H}_d$, $\mathbf{H}_r$ are unobserved, and jointly trained with the predictor $\hat{\mathbf{Y}}$.

Table 3: Results of MSE and VCF (all times $10^2$ for readability) on synthetic data of CIP with trade-off parameters depending on the chosen accuracy threshold.

|  | **VCF** $\times 10^2$ | **HSCIC** $\times 10^2$ |
|---|---|---|
| 90% accuracy | $3.14 \pm 0.92$ | $4.51 \pm 0.72$ |
| 70% accuracy | $3.01 \pm 0.80$ | $4.44 \pm 0.65$ |
| 1% accuracy | $2.91 \pm 0.92$ | $4.39 \pm 0.42$ |

counterfactual outcome is estimated as the average of these counterfactual predictions. In this experimental setting, we have $k = 100$ and $d = 1000$.

In the causal graph presented in Fig. 8, $\mathbf{A}$ includes the variables age and gender, $\mathbf{C}$ includes nationality and race, $\mathbf{M}$ marital status, $\mathbf{D}$ level of education, $\mathbf{R}$ the set of the working class, occupation, and hours per week and $\mathbf{Y}$ the income class. Compared to Chiappa & Pacchiano (2021), we include the race variable in the dataset as part of the baseline features $\mathbf{C}$. The loss function is the same as Eq. (1) but Binary Cross-Entropy loss ($\mathcal{L}_{\mathbf{BCE}}$) is used instead of Mean-Squared Error loss:

$$\mathcal{L}_{\text{CIP}}(\hat{\mathbf{Y}}) = \mathcal{L}_{\mathbf{BCE}}(\hat{\mathbf{Y}}) + \gamma \cdot \text{HSCIC}\left(\hat{\mathbf{Y}}, \{\text{age}, \text{gender}, \text{marital status}, \text{education}, \text{work}\} \middle| \mathbf{S}\right), \qquad (14)$$

where the set $\mathbf{S} = \{\text{Race}, \text{Nationality}\}$ blocks all the non-causal paths from $\mathbf{W} \cup \mathbf{A}$ to $\mathbf{Y}$. In this example we have $\mathbf{W} = \{\mathbf{C} \cup \mathbf{M} \cup \mathbf{D} \cup \mathbf{R}\}$. The results in Fig. 5 (right) refer to one run with conditioning set $\mathbf{S} = \{\text{Race}, \text{Nationality}\}$. The results correspond to 4 random seeds.

### G.6 Illustrating the choice of $\gamma$

In Section 3.4, we propose to choose $\gamma$ to obtain a maximal level of CI within a given tolerance on predictive performance. Here, we illustrate results from running the proposed procedure that dynamically selects $\gamma$, adjusted to different predefined accuracy thresholds in a classification setting. Specifically, the algorithm chooses the largest $\gamma$ value that yields an accuracy equal to or better than the threshold. As described the algorithm operates on $\gamma$ values on a logarithmic scale, thereby ensuring a fine-grained search over a wide range of potential trade-off points. Table 3 shows the found trade-offs for tolerated accuracies of 90%, 70%, and 1% in the same setting as Appendix G.1.

### G.7 Computational complexity and runtimes

In a dataset with $n = 1000$ data points from the setting discussed in Fig. 2, the average training time for one epoch without the regularization term is 0.003s and 1.112s with the regularization term. In these results, Adam optimizer with batch-size of 512 was used. By using smaller batch sizes, e.g. $n = 128$, the extra computational cost can further decrease. From a theoretical perspective, the estimation of the HSCIC requires kernel ridge regression (see Eq. (2) in our submission). In the high-dimensional image example, with a mini batch-size of 512, the average running time for an epoch with the regularization term is 64.03s and 34.01s without. Kernel ridge regression generally has a runtime that scales as $O(n^3)$ and memory requirements scaling like $O(n^2)$, where $n$ is the size of the dataset. However, these bounds can be significantly improved by using, i.e., random Fourier Features (see, i.e., Rahimi & Recht (2007); Avron et al. (2017)) as detailed in Appendix F. In short, by using random Fourier features, the resulting approximate kernel ridge regression estimator can be computed in a runtime of $O(ns^2)$ with $O(ns)$ memory. Here, $s$ is a parameter determining the accuracy of the approximation. In practice, $s$ can be set to be much smaller than the problem size, resulting in a dramatic speed-up. Other methods for efficient kernel computation include the popular Nystrom approximation (Drineas & Mahoney, 2005; Hsieh et al., 2014), and Memory-Efficient Kernel Approximation (MEKA) (Si et al., 2017). In this work, runtimes were still reasonable for all experimental settings, which is why we did not have to resort to these faster approximations.

## H Comparison with additional baselines

In this section, we compare CIP with additional baselines. These include Veitch et al. (2021) and different heuristic methods.

### H.1 Baseline experiments (Veitch et al., 2021)

We provide an experimental comparison against the method by Veitch et al. (2021). To this end, we consider the following data-generating mechanism for the causal structure (see Fig. 1(b)):

$$\mathbf{Z} \sim \mathcal{N}(0,1) \qquad \mathbf{A} = \sin(0.1\mathbf{Z}) + \varepsilon_{\mathbf{A}}$$

$$\mathbf{X} = \exp\left\{-\frac{1}{2}\mathbf{A}\right\}\sin(\mathbf{A}) + \frac{1}{10}\varepsilon_{\mathbf{X}}$$

$$\mathbf{Y} = \frac{1}{10}\exp\{-\mathbf{X}\} \cdot \sin(2\mathbf{XZ}) + \mathbf{AA} + \frac{1}{10}\varepsilon_{\mathbf{Y}},$$

where $\varepsilon_{\mathbf{X}}, \varepsilon_{\mathbf{A}} \overset{i.i.d}{\sim} \mathcal{N}(0,1)$ and $\varepsilon_{\mathbf{Y}} \overset{i.i.d}{\sim} \mathcal{N}(0,0.1)$. The data-generating mechanism of the anti-causal structure is the following (see Fig. 1(c)):

$$\mathbf{Z} \sim \mathcal{N}(0,1) \qquad \mathbf{A} = \frac{1}{5}\sin(\mathbf{Z}) + \varepsilon_{\mathbf{A}}$$

$$\mathbf{Y} = \frac{1}{10}\sin(\mathbf{Z}) + \varepsilon_{\mathbf{Y}}$$

$$\mathbf{X} = \mathbf{A} + \mathbf{Y} + \frac{1}{10}\varepsilon_{\mathbf{X}}$$

where $\varepsilon_{\mathbf{Y}}, \varepsilon_{\mathbf{A}} \overset{i.i.d}{\sim} \mathcal{N}(0,0.1)$ and $\varepsilon_{\mathbf{X}} \overset{i.i.d}{\sim} \mathcal{N}(0,1)$. We compare our method (CIP) against the method by Veitch et al. (2021) using different values for the trade-off parameter $\gamma$. In Fig. 1(b-c) the causal and anti-causal graphical settings proposed by Veitch et al. (2021) are presented. In both of these settings there is an unobserved confounder $\mathbf{Z}$ between $\mathbf{A}$ and $\mathbf{Y}$. The graphical assumptions outlined in Theorem 3.2 of the CIP are not met in the graphical structures under examination, as the confounding path is not effectively blocked by an observed variable ($\mathbf{Z}$ is unobserved). In light of this, it is assumed in our implementation that there is no unobserved confounder. In the graphical structure Fig. 1(b), CIP enforces $\text{HSIC}(\hat{\mathbf{Y}}, \mathbf{A} \cup \mathbf{X})$ to become small, gradually enforcing $\hat{\mathbf{Y}} \perp\!\!\!\perp \mathbf{A} \cup \mathbf{X}$. HSIC is the Hilbert-Schmidt Independence Criterion, which is commonly used to promote independence (see, i.e., Gretton et al. (2005); Fukumizu et al. (2007)). Veitch et al. (2021) enforces as independence criterion $\text{HSIC}(\hat{\mathbf{Y}}, \mathbf{A})$, which is implied by the independence enforced in CIP. In the anti-causal graphical setting presented in Fig. 1(c), the objective term used in CIP is $\text{HSCIC}(\hat{\mathbf{Y}}, \mathbf{A} \mid \mathbf{X})$, while in the method of Veitch et al. (2021) is $\text{HSCIC}(\hat{\mathbf{Y}}, \mathbf{A} \mid \mathbf{Y})$. In Table 4, the results of accuracy and VCF are presented.

In the experiments, the predictor $\hat{\mathbf{Y}}$ is a feed-forward neural network consisting of 8 hidden layers with 20 nodes each, connected with a rectified linear activation function (ReLU) and a linear final layer. Mini-batch size of 256 and the Adam optimizer with a learning rate of $10^{-4}$ for 500 epochs were used.

### H.2 Comparison baselines heuristic methods

We provide an experimental comparison of the proposed method (CIP) with some heuristic methods, specifically data-augmentation-based methods. We consider the same data-generating procedure and causal structure as presented in Appendix G.1. The heuristic methods considered are *data augmentation* and *causal-based data augmentation*. In the former, data augmentation is performed by generating $N = 50$ samples for every data-point by sampling new values of $\mathbf{A}$ as $a_1, ..., a_N \overset{i.i.d}{\sim} \mathbb{P}_{\mathbf{A}}$ and leaving $\mathbf{Z}, \mathbf{L}, \mathbf{Y}$ unchanged. Differently, in the latter *causal-based data augmentation* method, we also take into account the causal structure given by the known DAG. Indeed, when manipulating the variable $\mathbf{A}$, its descendants (in this example $\mathbf{L}$) will also change. In this experiment, a predictor for $\mathbf{L}$ as $\hat{\mathbf{L}} = f_\theta(\mathbf{A}, \mathbf{Z})$ is trained on 80% of the original dataset. In the data augmentation mechanism, for every data-point $\{a, x, z, y\}$, $N = 50$ samples are generated by sampling new values of $\mathbf{A}$ as $a_1, ..., a_N \overset{i.i.d}{\sim} \mathbb{P}_{\mathbf{A}}$, estimating the values of $\mathbf{L}$ as $x_1 = f_\theta(a_1, z), ..., x_N = f_\theta(a_N, z)$, while leaving the values of $\mathbf{Z}$ and $\mathbf{Y}$ unchanged. Heuristic methods such as data-augmentation methods

Table 4: **Results of the MSE, VCF of CIP and the baseline (Veitch et al., 2021)** applied to the causal and anti-causal structure in Fig. 1(b-c). Although the graphical assumptions are not satisfied, CIP shows an overall decrease of VCF in both of the graphical structures, performing on par with the baseline Veitch et al. (2021) in terms of accuracy and counterfactual invariance.

| | CIP | | Veitch et al. (2021) | |
|---|---|---|---|---|
| | **MSE** $\times 10^2$ | **VCF** | **MSE** $\times 10^2$ | **VCF** |
| $\gamma = 0.5$ | $4.58 \pm 0.31$ | $0.19 \pm 0.02$ | $4.50 \pm 0.40$ | $0.19 \pm 0.02$ |
| $\gamma = 1.0$ | $5.60 \pm 0.36$ | $0.18 \pm 0.01$ | $5.45 \pm 0.41$ | $0.18 \pm 0.02$ |

| | CIP | | Veitch et al. (2021) | |
|---|---|---|---|---|
| | **MSE** $\times 10^2$ | **VCF** | **MSE** $\times 10^2$ | **VCF** |
| $\gamma = 0.5$ | $1.16 \pm 0.01$ | $1.69 \pm 0.16$ | $1.01 \pm 0.01$ | $1.71 \pm 0.26$ |
| $\gamma = 1.0$ | $1.37 \pm 0.02$ | $1.48 \pm 0.19$ | $0.99 \pm 0.01$ | $1.88 \pm 0.28$ |

Table 5: Results of MSE and VCF (all times $10^2$ for readability) on synthetic data of CIP with trade-off parameters $\gamma = 0.5$ and $\gamma = 1$ with the heuristic methods *data augmentation* and *causal-based data augmentation* and *naive prediction*.

| | **VCF** $\times 10^3$ | **MSE** $\times 10^3$ |
|---|---|---|
| data augmentation | $3.12 \pm 0.16$ | $0.03 \pm 0.01$ |
| causal-based data augmentation | $3.04 \pm 0.16$ | $0.13 \pm 0.12$ |
| CIP ($\gamma = 0.5$) | $1.05 \pm 0.13$ | $1.64 \pm 0.22$ |
| CIP ($\gamma = 1.0$) | $0.35 \pm 0.19$ | $2.50 \pm 0.72$ |
| naive prediction (ignore $A$) | $9.01 \pm 0.02$ | $3.01 \pm 0.91$ |

do not theoretically guarantee to provide counterfactually invariant predictors. The results of an empirical comparison are shown in Table 5 with the average and standard deviations after 5 random seeds. It can be shown that these theoretical insights are supported by experimental results, as the VCF metric measure counterfactual invariance is lower in both of the two settings of the CIP ($\gamma = \frac{1}{2}$ and $\gamma = 1$).

A dataset of $n = 3000$ is used, along with $k = 500$ and $d = 500$. The architecture for predicting **L** and **Y** are feed-forward neural networks consisting of 8 hidden layers with 20 nodes each, connected with a rectified linear activation function (ReLU) and linear final layer. Mini-batch size of 256 and the Adam optimizer with a learning rate of $10^{-3}$ for 100 epochs were used.