# OpenReview forum: "Learning Counterfactually Invariant Predictors"
_TMLR — Accepted by TMLR_

### Review · Reviewer_8wSh · 2024-03-24

**Summary Of Contributions:**

The paper develops a graphical criteria such that the predictor is counterfactually invariant in terms of conditional independence in observational distribution. The authors then propose an estimation method based on kernel-based conditional dependence measure, and provide experiment results on simulated and real-world data.

**Audience:**

Yes

**Broader Impact Concerns:**

I am not aware of any broader impact concern.

**Claims And Evidence:**

Yes

**Requested Changes:**

- I would suggest moving the contributions part or at least provide a summary in the introduction. Currently, it may be unclear what the specific goal is after reading the introduction.
- What is the computational complexity and run time of the method?
- When going through the proofs, most of the theoretical results are based on existing results. Can the paper clarify which part of the theoretical results are new?

**Strengths And Weaknesses:**

Strengths:
- The paper is clearly written.
- The authors provide a detailed introduction to preliminaries and related works.
- The proposed method of counterfactually invariant prediction is sensible.
- The result appears to be sound, although I am not familiar with the literature to carefully go through the proof.

Weaknesses:
- The use of HSIC may limit the efficiency and scalability of the method.

I am not sufficiently familiar with the literature and existing works to list other major weaknesses, and am open to hearing thoughts from other reviewers.

---

> ### Author Response · Authors · 2024-05-10
> **Official response to Reviewer 8wSh**
>
> We thank the reviewer for the helpful comments and suggestions. We address the requested changes one by one.
>
> - **Moving contributions**: Thanks for the helpful suggestion. We added a short summary of the contributions earlier in the introduction. The more detailed overview also remained, since it references notions only described in the related work part.
> - **Computational cost**: We address this comment in the following ways: (a) we provide an additional section in the appendix (F.7) including the runtimes of some of our experiments with and without the HSCIC term (slowdowns between 2x and several 100x) as well as the computational complexity (dominated by O(n^3) of the kernel ridge regression). (b) Also in F.7 and the previous section E, we describe in detail how our computations may be sped up substantially by using random Fourier Features to approximate the kernel ridge regressions. (c) We added two sentences about the measured performance in the main paper (right before section 3.5) and pointers to the relevant sections in the appendix.
> - **Emphasizing novel theoretical contributions**: Thanks for mentioning this. All of the results in section A.4 are novel theoretical results. At the beginning of appendix A, we now provide a summary indicating which results are known and which are original. We also added a sentence in the main text to clarify our original theoretical contribution (right before Theorem 3.2).

---

### Review · Reviewer_iEPE · 2024-04-07

**Summary Of Contributions:**

This paper formalizes a notion of counterfactual invariance and establishes a sufficient condition for counterfactually invariant predictors via conditional independence. The paper then proposes a nonparametric framework to learn such predictors by using a kernel-based conditional independence measure as a regularization term, thereby striking a trade-off between predictive performance and counterfactual invariance.

**Audience:**

Yes

**Claims And Evidence:**

Yes

**Requested Changes:**

The paper can be strengthened by adding more discussions on the form of the learning framework, numerical optimization procedures, and optimality issues.

**Strengths And Weaknesses:**

Strengths:

The paper is well written and easy to follow. The methodology is motivated by important machine learning applications such as algorithmic fairness and robustness in image and text classifications.

Weaknesses:
1. The motivation for using kernel mean embeddings is not well discussed. What are the major advantages of the method over other nonparametric conditional independence measures? Also, how sensitive is the performance of the proposed method to the choice of kernel?
2. The optimization problem (1) does not seem very easy to solve. In particular, when an MLP is used as the learner, can the problem be solved efficiently? How was the optimization carried out in the experiments?
3. Instead of the penalized form of (1), one can consider a constrained form of the problem: minimize $\mathcal{L}(\hat{\mathbf{Y}})$ subject to $\mathrm{HSCIC}(\hat{\mathbf{Y}},\mathbf{A}\cup\mathbf{W}\mid\mathbf{S})\le\gamma$. The constrained form could be more appropriate for certain applications. For instance, in the study of algorithmic fairness, it would be desirable to treat counterfactual invariance as a requirement subject to a tolerance level rather than a property to trade-off.
4. Can the optimality of the predictor be studied in some way? Since prediction and counterfactual invariance are two fundamentally different goals to trade-off, it seems that the usual notion of optimality in terms of MSE or causality no longer applies. Is it possible to develop a concept of optimal predictors, say, optimal prediction among the class of nearly counterfactually invariant predictors?

---

> ### Author Response · Authors · 2024-05-10
> **Official response to Reviewer iEPE**
>
> We thank the reviewer for the helpful comments and suggestions. We address the requested changes and questions one by one.
>
> 1. **motivation for using kernel mean embeddings**: As far as we are aware most successfully deployed non-parametric conditional independence measures rely on KMEs. To the best of our knowledge, the main competing methods rely on conditional distance correlation metrics and conditional mutual information.
>
>    The conditional distance correlation (CDC) essentially requires a direct computation of the conditional correlation between pairwise distances of the random variables of interest. Although commonly used in economics, finance, and social sciences, CDC is impractical in our setting, where the conditioning set may be high-dimensional. Similarly, mutual information-based measures, relying on conditional Shannon entropy, require estimating the entropy of conditional distributions from finite samples, which is similarly intractable in high dimensions like our settings [1, 2].
>
>    The main advantage of our approach is that we do not need to estimate conditional covariances or entropies from samples. Instead, we use KMEs of conditional measures, for which strong convergence rates are known [3]. Our KMEs-based approach allows us to handle high-dimensional data and effective estimation from relatively few samples. Overall, kernel mean embeddings have a strong performance and reliability track record, particularly in complex data environments.
> 2. **choice of kernel**: Our theoretical analysis extends to any characteristic kernel. In practice, however, the choice of kernel determines the shape and width of the smoothing function applied to the data, affecting the smoothness of the estimated density and, consequently, the accuracy of the HSCIC estimate. The choice of kernel depends on the specific characteristics of the data. In practice, the RBF kernel used in our work has established itself as the default option in most settings. Due to the dependence on the data, any performance comparison and sensitivity analysis will necessarily be specific to the chosen dataset and not necessarily provide generalizable insights. How to choose a kernel (and its hyperparameters) in a data-driven fashion for kernel-based methods is an interesting and active area of research [4].
> 3. **optimization details**: Regarding the optimization of problem (1) using MLPs, in the implementation, we use the Adam optimizer. The specific parameters of these optimizers are detailed in the appendix within each experiment section. While there are theoretical results on stochastic approximation and when one may expect local stochastic optimization methods to converge to a local optimum, we cannot expect any theoretical guarantee of accurately solving the (global) optimization problem in (1)---just like virtually all deep-learning/gradient-based optimizations in machine learning have no theoretical (global) convergence guarantees. We conducted extensive testing as well as multiple random restarts under various configurations to ensure robustness and consistency in the obtained optima. These findings support the adequacy of achieving consistent local minima through our approach.
>
>      **constrained optimization instead of penalization**: Indeed, there are various ways one could think about this problem. We have described one potential mode of operation in the paragraph “The meaning of γ and how to choose it” in section 3.4. There, we suggest fixing a “tolerance level for the accuracy”. On the flip side, the suggestion of fixing a “CI violation tolerance” and aiming to achieve the best performance within that tolerance is also a desirable mode of operation. In our framework, these are really two sides of the same coin, as the parameter gamma traces out a Pareto frontier for this tradeoff. I.e., instead of performing the suggested search on gamma with a fixed accuracy/predictive performance tolerance, one can equally conduct this search for a target level of maximally tolerable violation of CI measured by HSCIC. We added a sentence about this mode of operation at the end of the paragraph on “The meaning of $\gamma$ and how to choose it”. Since in all our experiments, counterfactual invariance and predictive performance are indeed “working against each other”, we posit that a constrained optimization perspective with an inequality constraint on HSCIC would always land “on the boundary”, i.e., always achieve the maximally tolerated level of HSCIC. Hence, the solution of the constrained problem will coincide with the unconstrained one for the specific level of gamma that corresponds to the set level of HSCIC.

---

> > ### Author Response · Authors · 2024-05-10
> > **Official response to Reviewer iEPE (Part 2)**
> >
> > 4. **notions of optimality**: This is a very interesting and relevant question, to which we have no good answers at this point. Besides demonstrating Pareto-dominance over other competing methods **within a given data set and task**, we believe more universal optimality measures in such settings would be a valuable contribution to this and related tasks with competing objectives.
> >
> >
> > **References**:
> >
> > [1]: Wang X, Pan W, Hu W et al (2015) Conditional distance correlation. J Am Stat Assoc 110(512):1726–1734
> >
> > [2]: Lin, J. (1991). Divergence measures based on the Shannon entropy. IEEE Transactions on Information theory, 37(1), 145-151.
> >
> > [3]: Muandet, K., Fukumizu, K., Sriperumbudur, B., & Schölkopf, B. (2017). Kernel mean embedding of distributions: A review and beyond. Foundations and Trends® in Machine Learning, 10(1-2), 1-141.
> >
> > [4]: Simon-Gabriel, C. J., & Schölkopf, B. (2018). Kernel distribution embeddings: Universal kernels, characteristic kernels and kernel metrics on distributions. Journal of Machine Learning Research, 19(44), 1-29.

---

### Review · Reviewer_sMhK · 2024-05-06

**Summary Of Contributions:**

This paper reduces counterfactual invariance to conditional independence under unconfoundedness (conditioning on adjustment variables) and, importantly, a set of conditional variables W whose exogenous information is determined, given themselves and the treatment variable; the latter is achieved by assuming that each of W has a structural function that is injective in its exogenous noise. Based on the previous result, the paper proposes a learning approach (called CIP), adding a regularisation term that trades accuracy for the aforementioned conditional independence. Experiments on (semi)-synthetic datasets show the effectiveness of the approach.

**Audience:**

Yes

**Claims And Evidence:**

Yes

**Requested Changes:**

### Critical

The relationships of the current work to (Fawkes & Evans, 2023) should be clearer.

1. Besides D-CI, what are a.s.-CI and F-CI (Fawkes & Evans, 2023) is unclear. Your def of D-CI also includes an “a.s.” which is confusing, and I assume it is not the same as a.s.-CI. I also assume F-CI is not your D-CI applied to $\hat{Y}$ (if so, the eq of F-CI and D-CI is trivial and needs not to be proved).
2. The statement “On the contrary, the weaker notion of D-CI in Definition 2.2…” is confusing and incorrect. First, I believe you mean D-CI is weaker than a.s.-CI, but the “the weaker notion of D-CI in Definition 2.2” at first reads to me “our def 2.2 of D-CI is weaker than the D-CI defined in  (Fawkes & Evans, 2023)”. Second, whatever a.s.-CI means, I guess it shouldn’t be stronger than D-CI? Finally, and most importantly, I believe you can get Th 3.2 not because D-CI is stronger or weaker than a.s.-CI, but because you introduce the injectivity.
3. I think it is unfair to say (Fawkes & Evans, 2023) “*only* [prove] for special cases where the counterfactual distribution is identifiable”. Although I am not sure what is exactly assumed in (Fawkes & Evans, 2023), in which you could add an explanation, your injectivity assumption in essence says exogenous uncertainty, which is the core challenge of counterfactual inference, can be removed! This is in the similar spirit of assuming counterfactual identifiability and is not weaker.

### Good to have

The def of SCM allows a set of structural functions rather than a specific one; this deviates from the standard and is worth a caveat for inexperienced readers.

Def 3.1 and Th A.2 together are just the well-known backdoor criteria and this should be mentioned.

I am not sure why we need VCF in addition to the conditional indep itself to measure CI? In the Middle of Fig 2, you show they are strongly correlated, which is as expected.

In Fig 5, why the VCF of PSCF is not presented?

The proof of Th 3.2 is quite pretentious. Lemma A.5-7 are just different ways of saying “exogenous information of W is determined, given W themselves the treatment variable”. I guess we can find a short proof for Lemma A.7 directly.

I am not sure why you seem to stress the CIP eq1, by saying it is “our main contribution”, more than Th 3.2. For me, getting to eq1 from Th 3.2 is trivial, and eq1 is a trade-off (if we can ensure CI and at the same time prediction is great, I’d say this is more interesting than Th 3.2 alone, but surely this should be future work).

Of course, discussions of a couple of real-world examples demonstrating how the assumptions could be satisfied would be very interesting, but I guess this would be challenging.

### Minor

In “Even for granular W, there may be multiple units…”, I think “granular” should rather be “fine-grained”?

Lemma A.6, (y) → (t) in the last eq.

**Strengths And Weaknesses:**

### Strengths

Counterfactual invariance is an important but less studied problem in the machine learning community.

The development of the approach is solid and the theoretical analysis is rigorous.

### Weaknesses

Both the main Theorem 3.2 and the learning approach eq1 are quite straightforward. See Requests for details.

Consideration of real-world applicability is limited.

---

> ### Author Response · Authors · 2024-05-10
> **Official response to Reviewer sMhK**
>
> We thank the reviewer for the helpful comments and suggestions. We address the requested changes one by one.
>
> ## Critical
>
> Thanks a lot for carefully checking this part, we agree that our writing was not clear on the different (non)connections.
>
> 1. **a.s.-CI, D-CI, and F-CI**: We rewrote the paragraphs in Section 2.2. describing these different notions of CI. For the concrete questions: None of the notions defined in Fawkes & Evans, 2023 (D-CI, a.s.-CI, F-CI) coincides directly with ours, but our Definition 2.2 is most closely related to D-CI, but we do not enforce conditioning on the intervening variable. The “almost sure” in our definition indeed is not related to a.s.-CI, but the type of equality of distributions in a distributional CI notion. We have added a clarifying footnote on this in Def. 2.2.
> 2. **connection with D-CI**: We thank the reviewer for pointing this out. This sentence could indeed be confusing. D-CI is weaker than a.s.-CI (as defined in Fawkes & Evans, 2023), which they state in their Lemma 2.4. The sentence “On the contrary, …” aims to state that D-CI is weaker than a.s.-CI. For a.s.-CI it is impossible to infer CI from conditional independencies, whereas for D-CI this is actually possible under strong assumptions. Our notion of CI is most similar to D-CI and as Theorem 3.2 demonstrates, we can thus achieve CI from conditional independence. This would not be possible for a.s.-CI. We clarified this in our manuscript.
> 3. **counterfactual identifiability**: We agree that the injectivity assumption of our statement (Thm. 3.2) amounts to being able to remove exogenous uncertainty and thus is on similar footing as considering only identifiable counterfactual distributions. We clarified this fact and the connection to the statements in Fawkes & Evans 2023 before the “Contributions” paragraph in section 2.2.
>
> ## Good to have
>
> 1. **set of structural functions in def of SCM**: We believe this may be just unfortunate phrasing on our end: There is still just a single fixed structural equation (function) for each variable in our definition of an SCM. The (capital) F in the tuple is a **set of functions** (one for each endogenous variable in the SCM). Is this clear, or would you suggest changing the wording in the definition to disambiguate this?
> 2. **known definitions**: Thanks for the useful comment. We have worked on delineating our own theoretical contributions more clearly from existing definitions and results in the literature. At the beginning of Appendix A, we now provide a summary indicating which results are known and which are original. We also added the original statements and references directly in the header of known results, such as Theorem A.2 and common definitions. We also added a sentence in the main text to clarify our original theoretical contribution (right before Theorem 3.2).
> 3. **HSCIC and VCF**: We absolutely agree that for all practical purposes HSCIC is the relevant measure (also estimable from data). However, theoretically, only an exact value of HSCIC=0 implies counterfactual invariance (Corollary 3.5) and it is a-priori not clear whether “small values of HSCIC imply small values of counterfactual invariance”. Theoretically, we observe that the HSCIC is “continuous”, i.e., it converges to zero as the distributions converge weakly (see paragraph after Corollary 3.5). Hence, we believe that VCF as a measure of counterfactual invariance directly, still has merit, particularly because the scale and order of magnitude of HSCIC depend heavily on the data (dimensions) and it is harder to interpret whether a given value of HSCIC amounts to small or large violations of counterfactual invariance, whereas VCF is arguably a more interpretable and direct measure. One aspect of our experiments on VCF is to justify HSCIC as a practical measure of CI.
> 4. **VCF in PSCF**: Similarly to CF1, by construction, the VCF of PSCF is zero. Therefore, we omitted CF1 and PSCF from the figure for readability and only mention that their values are zero by design in the main text (Section 4.3).
> 5. **shorter proof of A.7**: With the benefit of hindsight, we agree that many of our proofs appear unnecessarily lengthy, given one has grasped the intuitive idea of “exogenous information being determined.” Admittedly, we only fully grasped this intuition after carefully going through the theory. In general, we strongly believe that spelling out proofs in detail (with explanations for all steps) is a worthwhile endeavor in machine learning papers, especially for readers without a deep intuition of structural causal models and counterfactuals (where it can quickly become hard to understand “which distribution one is talking about”). Hence, while we agree that our proofs can likely be compressed, we could not find a good way without assuming substantial background knowledge of the reader or skipping (relevant) details.

---

> > ### Author Response · Authors · 2024-05-10
> > **Official response to Reviewer sMhK  (Part 2)**
> >
> > ## Good to have (Part 2)
> >
> > 6. **main contribution**: Thanks for pointing this out. We agree with the reviewer’s reasoning of (1) essentially being a consequence of Theorem 3.2. With eq. (1) we aim to emphasize that this criterion can be effectively implemented in practice with a regularization learning approach. Hence, in our view, the main contribution is combining a solid theoretical foundation provided in Theorem 3.2 and the flexible, easy-to-use, practical learning framework (represented by eq. (1)).
> > 7. **real-world examples**: Indeed, assessing the validity of the assumptions in real-world settings is difficult. Generally, it is extremely difficult to convincingly argue even just for the validity of a causal DAG in real-world settings, where one can often conjecture plausible confounding mechanisms for any missing edge. As the reviewer anticipates, we found it similarly challenging to provide convincing examples that corroborate the “potential validity” of our assumptions. Ultimately, they remain untestable assumptions. Our empirical results demonstrate that our learning framework is effective even when the assumptions are violated, and our real-world experiments show a comparable invariance and accuracy trade-off trend as the simulated experiments. We believe this to be a better indicator of validity and robustness than isolated toy examples where assumptions may be satisfied.
> >
> > ## Minor
> >
> > Thanks a lot; we have fixed those.

---

> > > ### Comment · Reviewer_sMhK · 2024-05-14
> > > **Thank you**
> > >
> > > Most of the replies make sense and address my concerns. I'd like to follow up on some of them.
> > >
> > > Relationships to Fawkes & Evans, 2023. It is clearer now. I still suggest **quoting the mentioned defs and theoretical results in Appendix**. Also, side by side with the quotations, explanations could be added on the relationships between the D-CI and your Def 2.2, and how the D-CI and your Def 2.2 are weaker than as-CI and thus can be reduced to conditional indep, if these points are not obvious by looking at the quotations alone.
> > >
> > > Backdoor criteria. Sorry, I was not clear enough. I meant, Def 3.1 and Th A.2 might look foreign to some of the readers because they are the multi-variable version, while the phrase “backdoor criteria” would make an “aha” for them.
> > >
> > > Necessarily of VCF. Your explanation will strengthen the relevant parts of the paper and I strongly suggest adding it to Sec 3.5.
> > >
> > > CF1 and PSCF. Now, if I understand correctly, CF1 and PSCF are better in both prediction and CI on Adult? This is true, it should be mentioned in the text, and a tentative explanation of why so is highly desirable.
> > >
> > > Contribution. It would be nice to add this sentence to the paper: “the main contribution is combining a solid theoretical foundation provided in Theorem 3.2 and the flexible, easy-to-use, practical learning framework”.
> > >
> > > Real-world examples. I agree with you. And I suggest adding a similar explanation to the paper as a caveat for the practitioners, stressing they are recommended to do sanity checks similar to the experiments in this paper before they apply the method to their own data.

---

> ### Author Response · Authors · 2024-05-15
> **Follow up on reviewer's comments**
>
> We are happy that our replies addressed most of your concerns. We thank the reviewer for the helpful follow-up comments, we answer them one by one.
>
> **Relationships to Fawkes & Evans, 2023.** We are happy to read that this is clearer now. We added an additional appendix section (*Related work Fawkes & Evans, 2023* - App. A) including all of the CI definitions from Fawkes & Evans, 2023. We also included the relation between D-CI and Def 2.2, together with the restated theoretical results from Lemma 2.4 of Fawkes & Evans, 2023 comparing D-CI and a.s.-CI. In the previous revision, we had also added further comments on this comparison in the main paper (Section 2.2).
>
> **Backdoor criteria.** We added this to the revised version. Thanks for pointing this out.
>
> **Necessity of VCF.** We are happy our explanation clarified this. We added it to Sec. 3.5.
>
> **CF1 and PSCF.** Even if counterfactual invariance is fully satisfied in CF1 and PSCF (VCF=0), the considered CIP settings outperform PSCF in accuracy and performs on par to CF1. We explained this better in the revised version at the end of Sec. 4.3.
>
> **Contribution.** Thanks for the input, we added it to the introduction.
>
> **Real-world examples.** Following your suggestion, we now explain and emphasize this in the conclusions (Sec. 5).

---

> > ### Comment · Reviewer_sMhK · 2024-05-16
> > **Thank you, my concerns are addressed!**
> >
> > Thank you, my concerns are addressed!
> >
> > Two minor points:
> >
> > It is better to also mention "Backdoor criteria" in the main text.
> >
> > In the new Appendix A, you confused Fawkes & Evans, 2023 with Fawkes et al. (2022) at several places.

---

> ### Author Response · Authors · 2024-05-17
> **Response to reviewer**
>
> Thank you for the further comments, we are happy to read that the concerns have been addressed.
>
> Thank you also for noticing the citation mistake and for the suggestion. We included the recommendations in the revised version.

---

### Author Response · Authors · 2024-05-10
**Overview comment**

We thank the reviewers and the AC for working on this submission. We would like to summarize the main improvements of the revised version in response to the reviewers’ comments:

- We added a summary of our main contributions in the introduction section.
- We improved clarity of the Definition 2.2 of counterfactual invariance, together with its relations to D-CI, a.s.-CI and F-CI from Fawkes & Evans, 2023 as well as the strength of our injectivity assumption.
- We clarified the distinction between our theoretical contributions and previous definitions/results we build upon, both in the main paper and in the appendix.
- We added an entire section in the appendix on the computational complexity and runtimes in Appendix F.7 and provide further pointers to how the computational efficiency could be improved.

Overall, we appreciate all the reviewers' comments and feedback on the paper. We have addressed all the questions and improvement points, and we are happy to answer any other additional doubts.

---

### Decision · Action_Editor_qhnv · 2024-06-08

**Recommendation:** Accept as is

**Comment:**

I build my judgement on what I read and discussed with reviewers. In particular I agree that the manuscripts described a useful contribution  about the research subject of counterfactual invariance. The model agnostic which is developed and described together with a
computationally efficient learning framework suggest the paper to be of interest to the research community. I also agree that the application to fairness and robustness in image and text classifications give an additional motivation for acceptance. Furthermore, what described is sound. The works is theoretically solid even if maybe of a highly specialized are.

**Audience:**

Definitely Yes, me and colleagues from this research area find and will find what presented and discussed in the paper very interesting and useful for theoretical research as well as for fairness applications.

**Claims And Evidence:**

According to all the reviewers the paper, after the rebuttal phase, has to be judged very interesting and many comments from the reviewers confirm this. I liked and please very much supervising the review of this paper, thus I need to thank reviewers and authors who all were very collaborative and responsive. I suppirt the view of all reviewers that the paper claims are sufficiently supported by the given evidence under form of proofs and results of numerical experiments.